# Lineage-specific RUNX2 super-enhancer activates MYC and promotes the development of blastic plasmacytoid dendritic cell neoplasm

Sho Kubota [1], Kenji Tokunaga[2], Tomohiro Umezu [3], Takako Yokomizo-Nakano[1], Yuqi Sun[1], Motohiko Oshima[4,5], Kar Tong Tan[6], Henry Yang[6], Akinori Kanai[7], Eisaku Iwanaga[2], Norio Asou[8], Takahiro Maeda[9], Naomi Nakagata[10], Atsushi Iwama[4,5], Kazuma Ohyashiki[3], Motomi Osato[6,11,12] & Goro Sashida [1]

Blastic plasmacytoid dendritic cell neoplasm (BPDCN) is an aggressive subtype of acute leukemia, the cell of origin of which is considered to be precursors of plasmacytoid dendritic cells (pDCs). Since translocation (6;8)(p21;q24) is a recurrent anomaly for BPDCN, we demonstrate that a pDC-specific super-enhancer of *RUNX2* is associated with the *MYC* promoter due to t(6;8). RUNX2 ensures the expression of pDC-signature genes in leukemic cells, but also confers survival and proliferative properties in BPDCN cells. Furthermore, the pDC-specific *RUNX2* super-enhancer is hijacked to activate *MYC* in addition to *RUNX2* expression, thereby promoting the proliferation of BPDCN. We also demonstrate that the transduction of *MYC* and *RUNX2* is sufficient to initiate the transformation of BPDCN in mice lacking *Tet2* and *Tp53*, providing a model that accurately recapitulates the aggressive human disease and gives an insight into the molecular mechanisms underlying the pathogenesis of BPDCN.

[1] Laboratory of Transcriptional Regulation in Leukemogenesis, International Research Center for Medical Sciences (IRCMS), Kumamoto University, 2-2-1 Honjo, Chuo Ward, Kumamoto 860-0811, Japan. [2] Department of Hematology, Kumamoto University, 1-1-1 Honjo, Chuo Ward, Kumamoto 860-8556, Japan. [3] Department of Hematology, Tokyo Medical University, 6-7-1 Nishi-Shinjuku, Shinjuku, Tokyo 160-0023, Japan. [4] Department of Cellular and Molecular Medicine, Chiba University, 1-8-1 Inohana, Chuo Ward, Chiba 260-8670, Japan. [5] Division of Stem Cell and Molecular Medicine, Center for Stem Cell Biology and Regenerative Medicine, The Institute of Medical Science, The University of Tokyo, 4-6-1 Shirokanedai, Minato, Tokyo 108-8639, Japan. [6] Cancer Science Institute of Singapore, National University of Singapore, Singapore 119077, Singapore. [7] Department of Molecular Oncology, Research Institute for Radiation Biology and Medicine, Hiroshima University, Hiroshima 739-0046, Japan. [8] Department of Hematology, International Medical Center, Saitama Medical University, Saitama 350-1298, Japan. [9] Department of General Medicine, Nagasaki University, Graduate School of Biomedical Science, Nagasaki 852-8523, Japan. [10] Division of Reproductive Engineering, Center for Animal Resources and Development (CARD), Kumamoto University, 2-2-1 Honjo, Chuo Ward, Kumamoto 860-0811, Japan. [11] Laboratory of Runx Biology, International Research Center for Medical Sciences (IRCMS), Kumamoto University, 2-2-1 Honjo, Chuo Ward, Kumamoto 860-0811, Japan. [12] Center for Metabolic Regulation of Healthy Aging (CMHA), Kumamoto University, Chuo Ward, Kumamoto 860-0811, Japan. These authors contributed equally: Sho Kubota, Kenji Tokunaga. Correspondence and requests for materials should be addressed to M.O. (email: csimo@nus.edu.sg) or to G.S. (email: sashidag@kumamoto-u.ac.jp)

Blastic plasmacytoid dendritic cell neoplasm (BPDCN) is a rare and aggressive hematological malignancy, characteristic of skin lesions followed by hematopoietic organ dissemination, and is highly resistant to conventional chemotherapies, resulting in a poor outcome[1,2]. BPDCN was previously classified as blastic NK-cell lymphoma or CD4+CD56 +hematodermic neoplasm because of the expression of the NK cell marker, CD56; however, subsequent phenotypic (e.g. BDCA-2, IL3-Ra/CD123) and molecular studies identified plasmacytoid dendritic cells (pDCs) as a normal cellular counterpart of BPDCN[3]. Recent genetic studies identified the activation of the RAS signaling pathway, inactivation of tumor suppressor genes (e.g. TP53 and CDKN2A), and loss-of-function mutations in epigenetic modifiers (e.g. TET2)[4], which are commonly mutated in myeloid malignancies[5], whereas chromosomal translocation (6;8)(p21;q24) was exclusively observed in patients with BPDCN[6–8].

RUNX2, located on chromosome 6p21, is strongly expressed in pDCs and is also required for the differentiation and migration of pDCs through its regulation of the expression of pDC-signature genes[9,10]. CD34+stem/progenitor cells in BPDCN patients have been shown to have significantly higher expression levels of RUNX2 than acute myeloid leukemia (AML)[11]. BPDCN cells were recently reported to harbor super-enhancers of RUNX2 as well as TCF4, which are known to be critical for the development of pDCs[12]. The TCF4 transcription factor maintains pDC identity by positively regulating the expression of transcription factors, including RUNX2, which, in turn, regulate TCF4 during the differentiation of pDCs[9,13].

Super-enhancers are large clusters of enhancers that are densely occupied by transcription factors and mediators, such as BRD4, which drive the transcriptional activation of critical genes that define cell identity through their regulation of differentiation[14]. BRD4 is a member of the bromodomain and extraterminal (BET) subfamily, which binds to acetylated histones and facilitates transcriptional activation in normal as well as malignant cells[15,16]. BRD4 inhibitors, such as JQ1 and iBET, preferentially inhibit the function of super-enhancers, leading to the selective repression of potent oncogenes in various tumors[17–19]. The inhibition of BRD4 was shown to significantly abrogate the proliferative capacity of BPDCN, at least in part, due to the suppression of TCF4 transcription[12,20]. Thus, these lineage-survival transcription factors appear to utilize the activation of super-enhancers from precursors of and/or mature pDCs and confer transformation properties in BPDCN.

The function of MYC, located on chromosome 8q24, is critical for the development of various tumors[21,22], and the expression of MYC is activated by a gene amplification and the disrupted regulation of tissue-specific enhancers of MYC (e.g. colon cancer, T-ALL)[23–25]. Translocation-induced enhancer hijacking has been shown to activate the expression of oncogenes including MYC (e.g. IgH-MYC in lymphoma, TCR-LMO2 in T-ALL)[15,26]. Previous studies reported that t(6;8)(p21;q24) involved adjacent regions to MYC and RUNX2 in the cells of BPDCN patients;[7,8,27] however, the biological function of t(6;8) currently remains unclear. Based on these findings, we successfully demonstrated that t(6;8) juxtaposed the MYC and pDC-specific RUNX2 super-enhancer in BPDCN cells. The deletion of the mutant-allele super-enhancer of RUNX2 significantly reduced the expression of MYC and impaired the proliferative capacity of BPDCN cells, indicating that the pDC-specific RUNX2 super-enhancer activates the transcription of MYC. In addition, we investigated how MYC and RUNX2 promote the transformation of BPDCN in mice lacking Tet2 and Tp53.

We herein demonstrate that the pDC-specific RUNX2 super-enhancer is hijacked to activate expression of MYC via t(6;8) in BPDCN cells, and unveil the molecular mechanisms underlying the pathogenesis of BPDCN, which originates from a precursor of pDCs by utilizing a mouse model.

## Results

**Enhanced expression of MYC in BPDCN cells harboring t(6;8).** Since the super-enhancer of RUNX2 has been detected in BPDCN cells harboring t(6;8), which involves a region adjacent to MYC[12,27], we initially examined the expression levels of RUNX2 and MYC in leukemic cells harboring t(6;8) in patients and the cell line, CAL-1, which has a loss-of-function mutation in TET2 (Supplementary Fig. 1)[6]. The expression levels of RUNX2 were significantly higher in BPDCN cells than in AML cells and U2OS and Saos2 osteosarcoma cells as Saos2 has higher expression level of RUNX2 than normal counterpart cells and promotes the cell growth[28], while the expression levels of RUNX2 were lower in BPDCN cells than in mature pDCs isolated from healthy donors (Fig. 1a). We found that BPDCN cells had markedly higher MYC expression levels among these malignant cells, whereas normal pDCs only negligibly expressed MYC (Fig. 1b). Furthermore, CAL-1 cells strongly expressed the MYC and RUNX2 proteins (Fig. 1c). Notably, a recent study reported that 22 out of 118 BPDCN patients harbored t(6;8) and enhanced expression of MYC, compared to BPDCN cells without t(6;8)[8]. Therefore, the expression of MYC are increased in BPDCN cells harboring t (6;8).

In order to identify the breakpoint of t(6;8)(p21;q24), we performed fluorescent in situ hybridization (FISH) utilizing three distinct probes against either the 5′ region of MYC (8q24), 1.7-Mb 3′ downstream region of MYC (8q24), which contains blood cell-specific MYC enhancers in normal cells and AML cells[24,29,30], or a coding region of the RUNX2 gene (6p21) in a patient and CAL-1 cells (Fig. 1d). We found an associated signal between RUNX2 and the 3′ enhancer region of MYC on derivative chromosome 6 (der(6)), and a single signal of the 5′ region of MYC on der(8) in BPDCN cells (Fig. 1e, and Supplementary Fig. 2). Consistent with the finding of a fusion point between the downstream region of MYC and upstream region of RUNX2 in other patients (Fig. 1d)[7,8], we performed whole-genome sequencing of CAL-1 cells, and identified a fusion point of chromosome translocation between chromosome 8:128,817,726 (hg19) in 69 kilobases (kb) downstream of MYC and chromosome 6: 44,657,458 (hg19) (Fig. 1d), which was 58 kb centromeric to a long and clustered enhancer of RUNX2 (791 kb upstream of RUNX2) defined by chromatin immunoprecipitation (ChIP) sequencing using anti-H3K27ac or anti-BRD4 antibodies (Fig. 1f)[12]. We analyzed the H3K27ac peak rank order for super-enhancers using the ROSE algorithm[15,16], and confirmed super-enhancers of RUNX2 and TCF4 in CAL-1 cells (Supplementary Fig. 3), as previously reported[12], while we found small and scattered peaks of H3K27ac and BRD4 in blood cell-specific MYC enhancers also known as BENC (blood enhancer cluster; Supplementary Fig. 4). Murine pDCs contained a long and clustered enhancer of Runx2 (543 kb upstream of Runx2) equivalent to that in humans, whereas multipotent progenitors (MPPs), common dendritic cell progenitors (CDPs), and conventional dendritic cells (DCs) did not (Supplementary Fig. 5)[31], suggesting that BPDCN cells utilize the super enhancer of RUNX2 (seRUNX2 -791kb), which is activated in normal pDCs. In order to clarify whether the super-enhancer of RUNX2 was physically associated with the MYC promoter on der(8), we performed a chromatin configuration capture (3 C) analysis and a target sequencing to detect the expected chromosomes 6 and 8 regions using CAL-1 cells and Jurkat T-ALL cells lacking t(6;8) (Fig. 1g and Supplementary Fig. 6). The results revealed that the

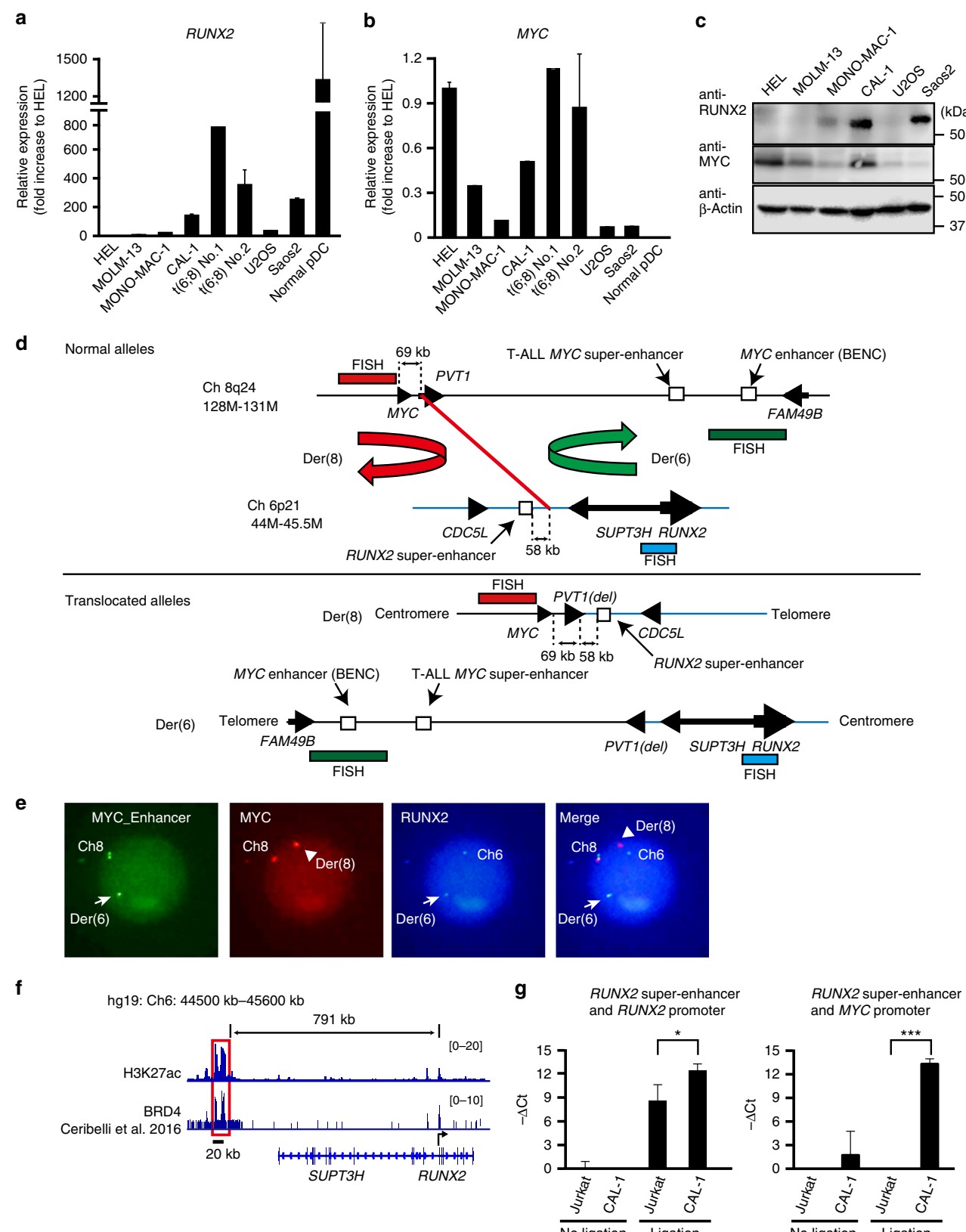

super-enhancer of *RUNX2* was significantly associated with the promoter of *MYC* and the promoter of *RUNX2* in CAL-1 cells, but not in Jurkat cells showing undetectable RUNX2 protein level (see below). These results indicate that BPDCN cells enhanced the expression of *MYC* following a juxtaposition of *MYC* and the pDC-specific seRUNX2 (-791kb) due to the translocation.

**MYC and RUNX2 collaborate to promote the tumor cell growth**. Based on the enhanced expression of MYC in BPDCN cells harboring t(6;8), we examined whether MYC and/or RUNX2 promoted the cell growth of BPDCN in vitro. We knocked down the expression of *MYC* using two distinct *MYC*-directed shRNA vectors in CAL-1 cells and found that both *MYC*-directed

**Fig. 1** Enhanced expression of MYC in BPDCN cells harboring t(6;8). **a** Expression levels of *RUNX2* mRNA in acute leukemia cell lines (HEL, MOLM13, and MONO-MAC1), CAL-1, bone marrow mononuclear cells harboring t(6;8) isolated from two patients (#1 and #2), and osteosarcoma cell lines (Saos2, U2OS) examined by quantitative RT-PCR (q-PCR) compared to those in normal pDCs isolated from healthy donors (*n* = 5). **b** Expression levels of *MYC* mRNA examined by q-PCR in the same cells described in Fig. 1a. **c** Expression levels of RUNX2 and MYC proteins in acute leukemia cell lines (HEL, MOLM13, and MONO-MAC1), CAL-1, and two osteosarcoma cell lines (Saos2, U2OS). Levels of β-Actin were used as loading controls. **d** Maps showing chromosomal regions of human 8q24 (129M-131M) (black line) and 6p21 (44M-47M) (blue line) (upper panel) and derivative chromosome regions of Der (6) and Der(8) (lower panel). Red line indicates a fusion point of t(6;8)(p21;q24) in CAL-1 cells identified by whole genome sequencing in this study. Red, green, and aqua bars indicate the target regions of FISH probes in Fig. 1e. **e** Association between the 3′ enhancer region of MYC (green signals) and the RUNX2 gene (aqua signals) observed in Der(6) in a patient (#2) assessed by FISH. Arrows and arrow heads show Der(6) and Der(8), respectively. **f** CAL-1 cells showing a long and clustered enhancer of RUNX2 (hg19: 44500kb−45600kb) assessed by ChIP sequencing utilizing either an anti-H3K27ac in this study or anti-BRD4 antibody[12]. **g** CAL-1 cells showing a significant association between the super enhancer of *RUNX2* and the promoter of either *MYC* or *RUNX2* assessed by a 3C-qPCR assay. DNA ligase non-treated samples were used as negative controls. Data are representative of three independent experiments. **a**, **b**, **g** Bars show the mean±SD, *\*p* < 0.05, and *\*\*\*p* < 0.001 by the Student's *t*-test

shRNA-transduced cells had significantly lower expression levels of *MYC* mRNA and protein than control vector-transduced cells (Fig. 2a, b). Given the potent oncogenic functions of MYC in various tumors, colony formation capacities were significantly weaker in *MYC* knocked down (KD) cells than in control vector-transduced cells (Fig. 2c). Thus, MYC functioned as an oncogene and enhanced the clonogenic capacity of BPDCN cells.

We also transduced two distinct *RUNX2*-directed shRNA vectors into CAL-1 cells, and confirmed that the mRNA and protein expression levels of RUNX2 were significantly lower in *RUNX2*-directed shRNA-transduced cells than in control vector-transduced cells (Fig. 2d, e). The number of *RUNX2*-directed shRNA-transduced cells was significantly lower than that of control vector-transduced cells in an in vitro liquid culture, and colony formation capacities were weaker in the former than in the latter (Fig. 2f, g). Consistent with the impaired cell growth capacities of *RUNX2* KD cells, *RUNX2* KD cells showed significantly lower BrdU incorporation than did the control cells (Fig. 2h). Furthermore, *RUNX2* KD cells had significantly more Annexin V-positive cells than control vector-transduced cells (Fig. 2i), indicating that RUNX2 enhanced proliferative capacities, but also suppressed inappropriate apoptosis in BPDCN cells. In addition, when we knocked down the expression of both *MYC* and *RUNX2* in CAL-1 cells, *MYC* and *RUNX2*-double KD cells showing significantly weaker clonogenic capacity than either single KD cells (Fig. 2j). Therefore, MYC and RUNX2 collaborate to promote the cell growth capacity of BPDCN in vitro.

**RUNX2 functions as a lineage-survival factor in BPDCN.** Since RUNX2 is required for the differentiation of pDCs[9], in order to clarify the mechanisms by which RUNX2 promotes the cell growth of BPDCN, we analyzed the gene expression profiles of two distinct *RUNX2* KD CAL-1 cells and control CAL-1 cells using a microarray analysis. These two *RUNX2* KD cells showed significant overlapping of genes that were either up- or down-regulated in comparison with the control cells (Fig. 3a). Up-regulated and down-regulated genes were listed in Supplementary Data 1. A gene set enrichment analysis (GSEA) revealed that *RUNX2* KD cells showed the negatively enriched expression of pDC signature genes, which was defined by genes expressed in pDCs (HLA-DR+CD123+CD11c−) compared with myeloid dendritic cells (mDCs) (HLA-DR+CD123−CD11c+) in humans (Fig. 3b)[32]. Quantitative RT-PCR confirmed that *RUNX2* KD cells had significantly lower expression levels of pDC-signature genes, such as *TCF4*, *TLR7*, *TLR9*, and *IL-3RA*, than control cells (Fig. 3c), indicating that RUNX2 maintained pDC-signature gene expression in BPDCN cells. Furthermore, *RUNX2* KD cells had significantly decreased expression levels of the BPDCN-characteristic surface markers, the CD56 and CD123 proteins (Fig. 3d), which was accompanied by impaired proliferation and

enhanced apoptosis upon the knockdown of *RUNX2* (Fig. 2h, i). Our results also demonstrated that *RUNX2* KD cells had significantly decreased expression levels of canonical MYC target genes (Fig. 3e) accompanied with the decreased levels of *MYC* mRNA expression (Fig. 3f), indicating that RUNX2 facilitated the activation of MYC targets in this context. We found that ectopic expression of MYC partially rescued the reduced colony formation capacity in *RUNX2* KD CAL-1 cells (Supplementary Fig. 7); which is consistent with the findings that RUNX2 required MYC to promote the survival of cancer cells[28,33]. These results indicate that RUNX2 ensures the expression of pDC-signature genes, but also confers survival and proliferative properties in leukemic cells, in part, due to the activated function of MYC.

**RUNX2 activates the function of enhancers in BPDCN.** Since the super-enhancer of *RUNX2* appeared to promote the development of BPDCN by activating the expression of *MYC* and *RUNX2*, we examined whether the BRD4 inhibitor JQ1 abrogated the proliferation of CAL-1 cells in vitro. As expected, JQ1-treated CAL-1 cells had significantly lower expression levels of *MYC* and *RUNX2* mRNA than DMSO-treated CAL-1 cells (Fig. 4a). We also confirmed that JQ1 markedly reduced the expression levels of the RUNX2 and MYC proteins in CAL-1 cells, while Jurkat T-ALL cells showed a mild change in MYC protein levels after the JQ1 treatment (Fig. 4b). Consistent with the reduced expression levels of these oncogenes, our results revealed that CAL-1 cells showed higher sensitivity to JQ1 in a dose-dependent manner, whereas Jurkat cells were resistant to this treatment (Fig. 4c). In addition, the colony formation capacities of CAL-1 cells were abolished by the treatment with JQ1 (Fig. 4d). The JQ1 treatment modestly decreased level of H3K27ac expression in CAL-1 cells (Fig. 4e), and H3K27ac ChIP sequencing revealed that H3K27ac modification levels were reduced at the region of *RUNX2* super-enhancer after the JQ1 treatment (Fig. 4f). Thus, the inhibition of BRD4 significantly impaired the growth of BPDCN cells following the reduced expression of *MYC* and *RUNX2*.

In order to clarify if RUNX2 and MYC were responsible for the formation of BPDCN due to activating the function of enhancers, we examined whether the ectopic expression of *RUNX2* and/or *MYC* rescued impaired cell growth after the JQ1 treatment. We confirmed the successful transduction of RUNX2 and/or MYC in CAL-1 cells (Fig. 4g). We found that neither RUNX2 alone, MYC alone, nor RUNX2 plus MYC overexpression promoted significant changes in growth capacities without the JQ1 treatment (Fig. 4h). While the ectopic expression of MYC alone caused slight increases in the growth of CAL-1 cells after the JQ1 treatment, RUNX2 alone and RUNX2 plus MYC overexpression significantly sustained the numbers of CAL-1 cells after the JQ1 treatment (Fig. 4h), indicating that RUNX2 plays a pivotal role in transcriptional networks for survival upon the insult of

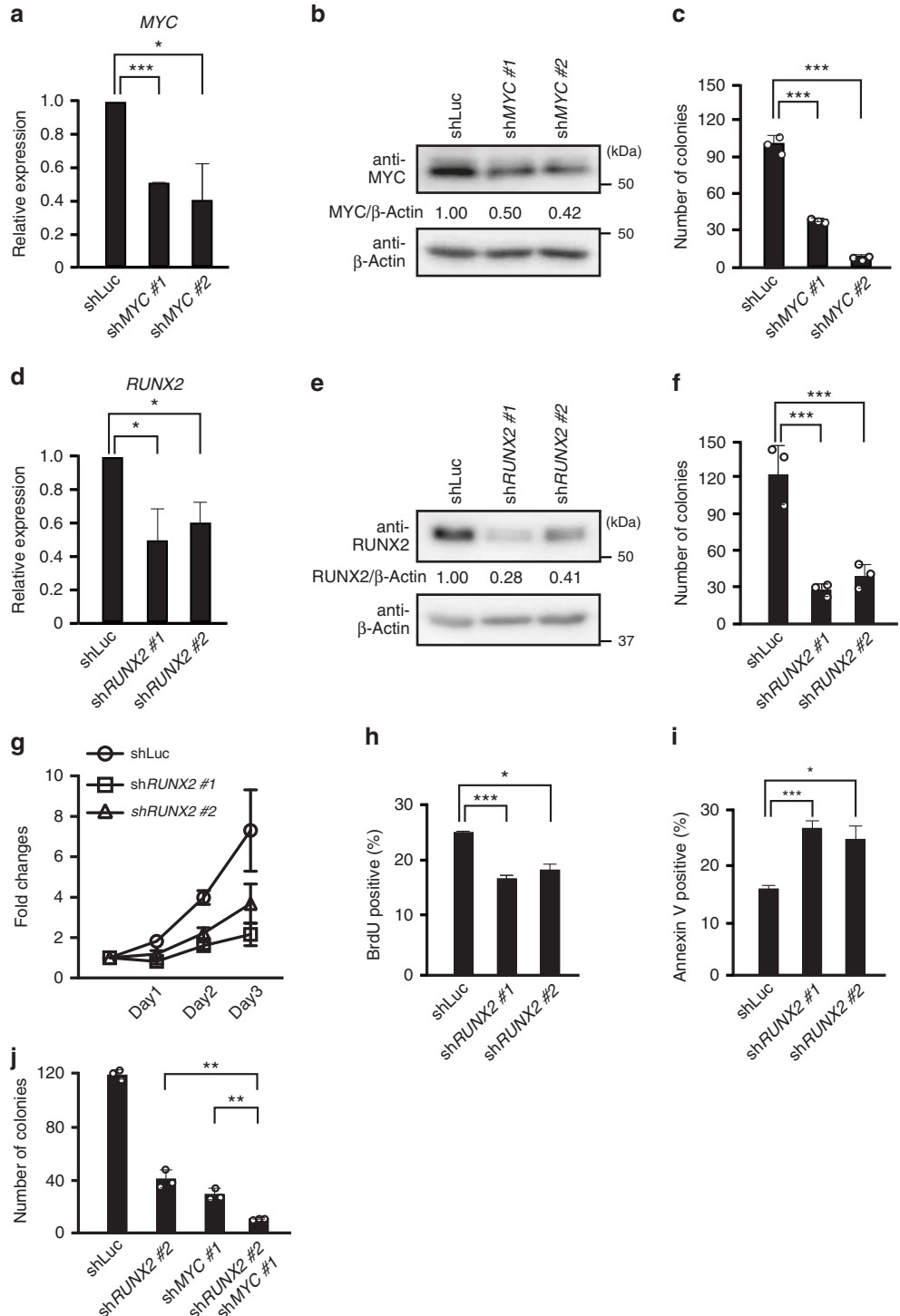

**Fig. 2** MYC and RUNX2 collaborate to promote the formation of BPDCN cells. **a** Expression levels of *MYC* mRNA in CAL-1 cells transduced with two distinct MYC-directed shRNA vectors and a control luciferase-directed shRNA vector. **b** Expression levels of the MYC protein in *MYC* KD CAL-1 cells. Levels of β-Actin were used as loading controls. **c** Impaired colony formation capacities in *MYC* KD CAL-1 cells from those in control cells (*n* = 3). **d** Expression levels of *RUNX2* mRNA in CAL-1 cells transduced with two distinct *RUNX2*-directed shRNA vectors and a control luciferase-directed shRNA vector. **e** Expression levels of the RUNX2 protein in *RUNX2* KD CAL-1 cells. Levels of β-Actin were used as loading controls. **f** Impaired colony formation capacities in *RUNX2* KD CAL-1 cells from those in control cells (*n* = 3). **g** Decreased cell growth in *RUNX2* KD CAL-1 cells (squares and triangles) under in vitro liquid culture conditions from that in control CAL-1 cells (circles) (*n* = 3). **h** Reduced proliferative capacities in *RUNX2* KD CAL-1 cells assessed by BrdU incorporation (*n* = 3). **i** Enhanced apoptosis in *RUNX2* KD CAL-1 cells over that in control CAL-1 cells (*n* = 3). **j** Reduced colony formation capacity in *MYC* and *RUNX2* double KD CAL-1 cells from those in single KD cells (*n* = 3). **a**, **c**, **d**, **f**, **g–j** Bars show the mean±SD, *$p < 0.05$, **$p < 0.01$, and ***$p < 0.001$ by the Student's *t*-test. Data are representative of 2–3 independent experiments

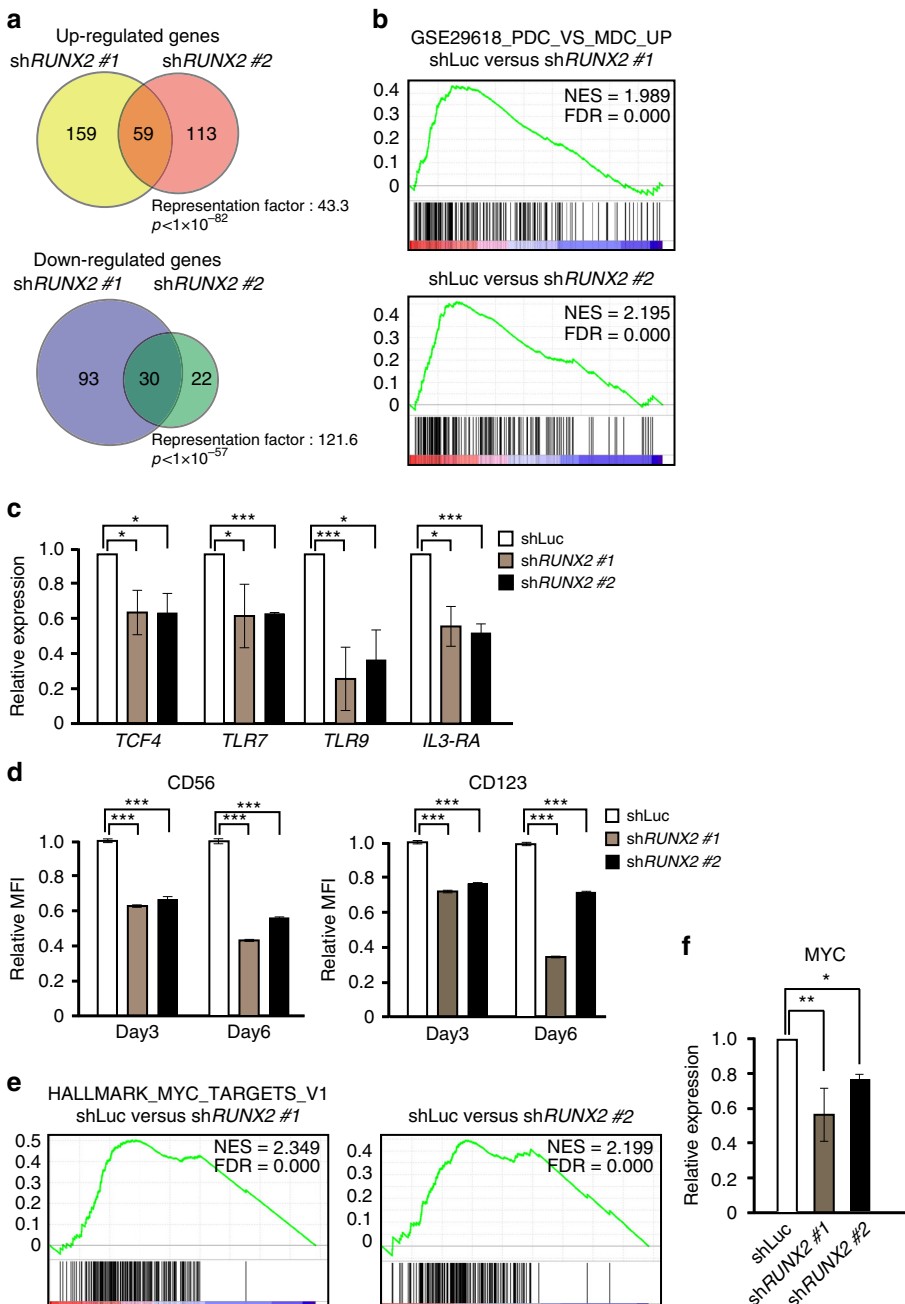

**Fig. 3** RUNX2 functions as a lineage-specific transcription factor in BPDCN. **a** Venn diagrams of significant overlaps in up-regulated and down-regulated genes between two distinct *RUNX2* KD CAL-1 cells relative to control CAL-1 cells. **b** GSEA for pDC signature genes, which was defined by up-regulated genes in normal pDCs versus mDCs (GSE29618), in two *RUNX2* KD CAL-1 cells relative to control CAL-1 cells. **c** q-PCR showing the reduced expression of pDC signature genes, such as *TCF4*, *TLR7*, *TLR9*, and *IL3-RA*, in *RUNX2* KD CAL-1 cells (gray and black bars). **d** Decreased mean fluorescence intensities (MFI) of CD56 and CD123/IL3-RA expression in *RUNX2* KD CAL-1 cells (gray and black bars) 3 and 6 days after transduction examined by FACS ($n = 3$). **e** GESA for canonical MYC target genes (Hallmark MYC targets V1) in two *RUNX2* KD CAL-1 cells relative to control CAL-1 cells. **f** Expression levels of *MYC* mRNA examined by q-PCR in *RUNX2* KD CAL-1 cells (gray and black bars) and control CAL-1 cells (opened bar). **c**, **d**, **f** Bars show the mean±SD, $*p < 0.05$, $**p < 0.01$, and $***p < 0.001$ by the Student's *t*-test. Data are representative of three independent experiments

BRD4 inhibition. Indeed, RUNX2 ChIP sequencing revealed that RUNX2 were significantly enriched in the RUNX2 super-enhancer region showing at least 5 peaks (Fig. 4f), and RUNX2-binding sites were markedly overlapped with super-enhancers rather than conventional enhancers (Supplementary Fig. 8). In addition, a motif enrichment analysis on DNA sequences of H3K27ac ChIP sequencing revealed that super-enhancers enriched the binding sequence of RUNX2 significantly more than conventional enhancers in CAL-1 cells (Supplementary

Fig. 9). Thus, RUNX2 and MYC promote the proliferation of BPDCN, at least in part, due to the RUNX2-dependent direct activation of super-enhancers, which is reversed by the inhibition of BRD4.

**RUNX2 super-enhancer is hijacked to activate MYC expression.** In order to clarify whether the *RUNX2* super-enhancer directly activated the expression of *MYC* in a *cis* manner, we

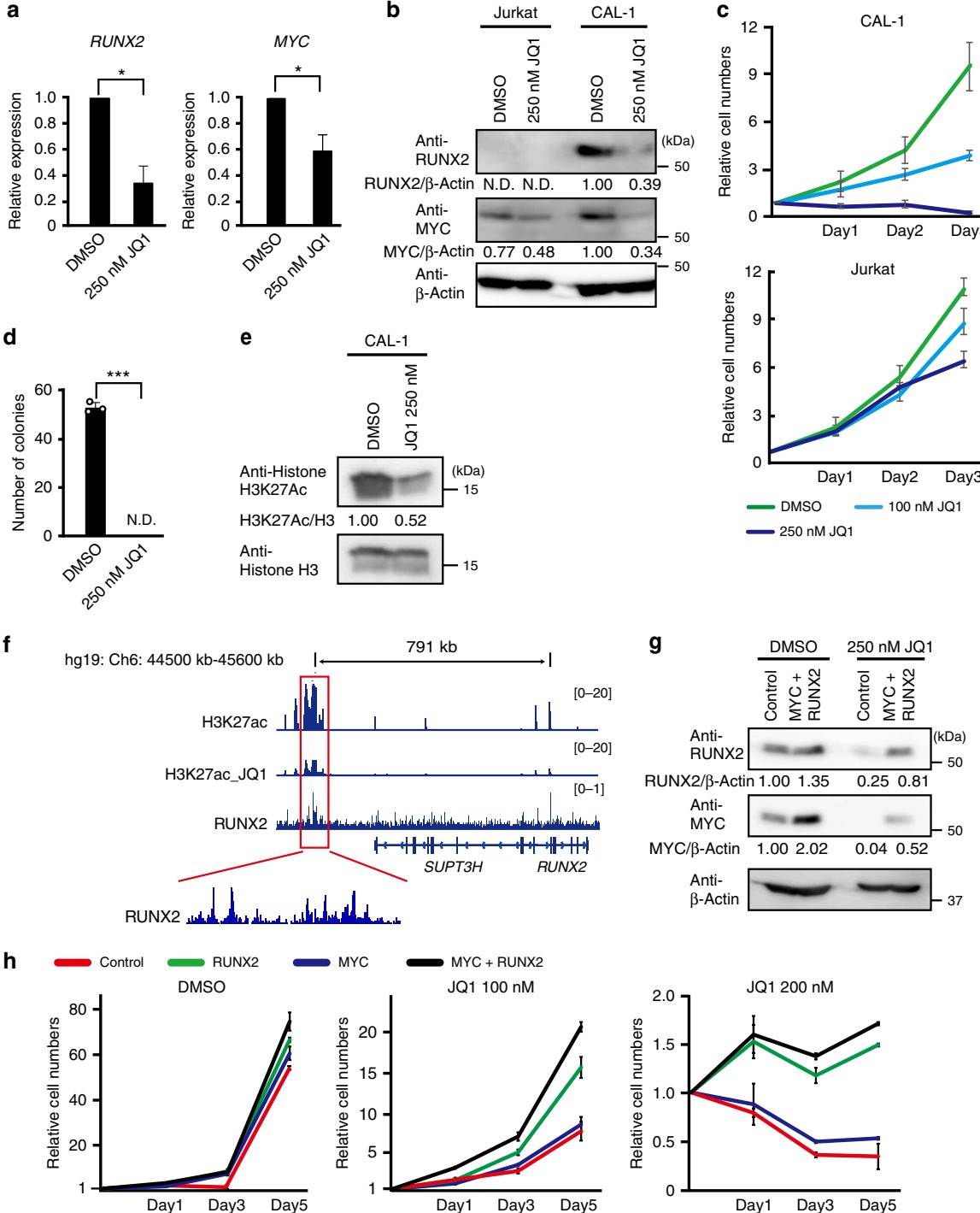

**Fig. 4** MYC and RUNX2 promote the formation of BPDCN by activating the function of enhancers. **a** Expression levels of *RUNX2* and *MYC* mRNA examined by q-PCR in CAL-1 cells 1 day after the 250 nM JQ1 treatment ($n = 3$). **b** CAL-1 cells showing reduced levels of the RUNX2 and MYC proteins 1 day after the 250 nM JQ1 treatment. Levels of β-Actin were used as loading controls. **c** Decreased cell growth capacities in CAL-1 cells after the JQ1 treatment at 100 nM (light blue line) and 250 nM (blue line) from that in cells treated with DMSO (green line), but not in Jurkat cells ($n = 3$). **d** Impaired colony formation capacities of CAL-1 cells treated with 250 nM JQ1 from that in cells treated with DMSO ($n = 3$). **e** Decreased level of H3K27ac expression in CAL-1 cells 1 day after the 250 nM JQ1 treatment. **f** RUNX2 binding to the super-enhancer of *RUNX2* that reduced level of H3K27ac post JQ1 treatment in CAL-1 cells assessed by H3K27ac- and RUNX2-ChIP sequencing. **g** Successful transduction of RUNX2 and MYC in CAL-1 cells with or without the JQ1 treatment at 250 nM examined by western blotting. Levels of β-Actin were used as loading controls. **h** RUNX2 alone and RUNX2 plus MYC transductions restoring the impaired cell growth post JQ1 treatment. Empty controls (red lines), RUNX2 alone (green lines), MYC alone (blue lines), and RUNX2 plus MYC vectors (black lines) were transduced in CAL-1 cells, which were then treated with JQ1 at 100 nM and 200 nM under liquid culture conditions ($n = 3$). **a**, **c**, **d**, **h** Bars show the mean±SD, *$p < 0.05$ and ***$p < 0.001$ by the Student's *t*-test. Data are representative of 2–3 independent experiments

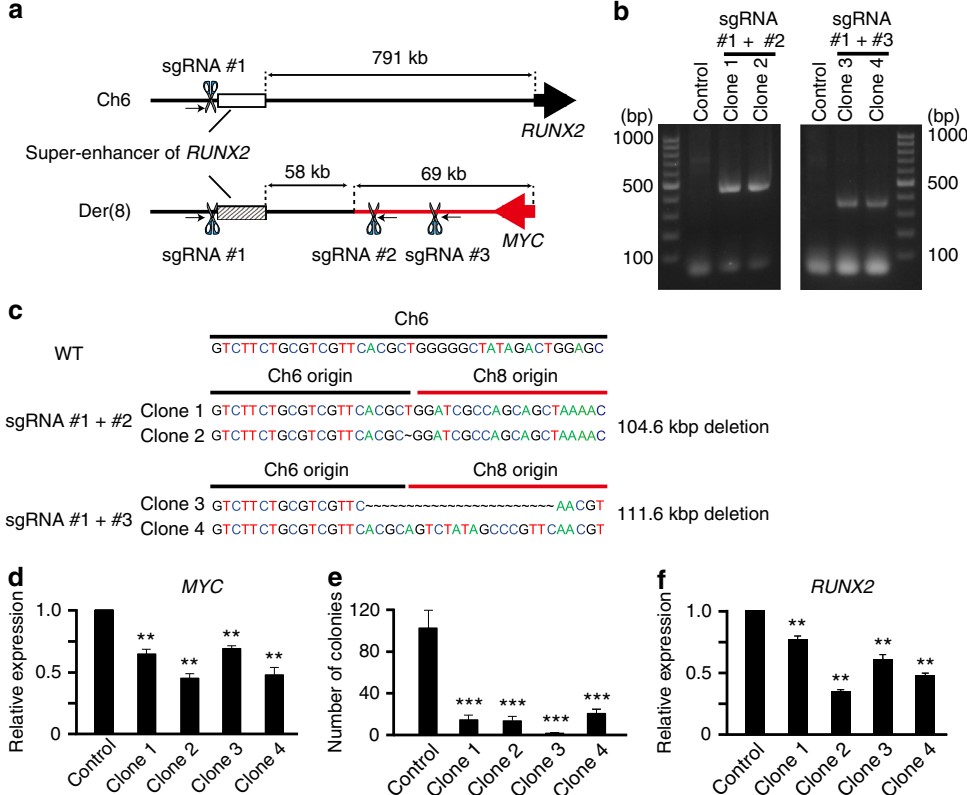

**Fig. 5** RUNX2 super-enhancer-driven MYC expression promotes the formation of BPDCN. **a** Schematic illustration of the *RUNX2* super-enhancers on chromosome 6 (se*RUNX2* (-791kb); a back line) and derivative chromosome 8 (se*RUNX2*der8; a black and red fused line). sgRNA vectors were directed a distal region of the super-enhancer (sgRNA #1; chr6: 44,557,984-44,558,006) and two distinct downstream regions of *MYC* (sgRNA #2; chr8:128,812,545-128,812,567 and sgRNA #3; chr8:128,805,565-128,805,587). Arrows indicate primers for genomic PCR utilized in Fig. 5b. **b** Detecting successful deletions of the se*RUNX2*der8 by genomic PCR using primers on distinct chromosomes of 6 and 8. **c** Sequences of se*RUNX2*der8-deleted clones elucidated by Sanger sequencing and sizes of deletion in those clones. **d** Expression levels of *MYC* mRNA in se*RUNX2*der8-deleted CAL-1 clones examined by q-PCR. **e** Impaired colony formation capacities in se*RUNX2*der8-deleted CAL-1 clones from those in control cells (*n* = 3). **f** Expression levels of *RUNX2* mRNA in se*RUNX2*der8-deleted CAL-1 clones examined by q-PCR. **d**–**f** Bars show the mean±SD, \*\*p < 0.01 and \*\*\*p < 0.001 by the Student's *t*-test

attempted to genetically delete the mutant-allele super-enhancer of *RUNX2* on der(8) (se*RUNX2*der8), but not that on chromosome 6 (se*RUNX2* (−791kb)), in BPDCN cells using a CRISPR-Cas9 system (Figs. 1d and 5a). We generated sgRNA-CRISPR-Cas9 vectors targeted for the distal region of the *RUNX2* super-enhancer on chromosome 6 (sgRNA #1) and two distinct downstream regions of the *MYC* gene on chromosome 8 (sgRNA #2 and sgRNA #3; Fig. 5a), infected vectors into CAL-1 cells, and then selected single cell clones. We successfully established four distinct clones of CAL-1 cells from two sgRNA vectors (#1 plus #2, and #1 plus #3), and confirmed a deletion of se*RUNX2*der8 by amplification with genomic PCR utilizing primers beyond the fusion point between chromosomes 6 and 8 (Fig. 5b). The Sanger sequences of PCR products revealed the expected 104 kb or 111 kb deletion including the se*RUNX2*der8 in these clones (Fig. 5c). After establishing single cell clones, we examined the expression levels of *MYC* using quantitative PCR, and found that the expression levels of *MYC* were significantly lower in se*RUNX2*der8-deleted clones than in control-vector transduced cells (Fig. 5d). All se*RUNX2*-mut-deleted clones showed markedly impaired colony formation capacities from those in control CAL-1 cells (Fig. 5e). We noted that these clones showed between 20 to 60% reductions in *RUNX2* expression levels (Fig. 5f), suggesting that MYC positively regulates the expression of *RUNX2* in a direct and/or an indirect manner. While the reduced expression of *MYC* and *RUNX2* may further inhibit colony formation capacities of

these clones, the mutant-allele super-enhancer of *RUNX2* directly activates the expression of *MYC* to promote the formation of BPDCN.

**Transduction of MYC promotes the in vitro colony formation.** Since MYC and RUNX2 promote the formation of BPDCN due to the *RUNX2* super-enhancer in vitro, we examined whether the over-expression of *MYC* and *RUNX2* was sufficient for the initiation of BPDCN in the absence of *Tet2* and *Tp53*, because loss-of-function mutations in *TET2* and *TP53* are the most frequently detected in patients with BPDCN regardless the presence of t(6;8)[4]. In order to achieve this, we generated *Tet2^{wt/wt}*; *Tp53^{wt/wt}*;*Cre-ERT2* and *Tet2^{flox/flox}*;*Tp53^{flox/flox}*;*Cre-ERT2* compound mice and deleted *Tet2* and *Tp53* via an injection of tamoxifen at 8 weeks old, hereafter referred to as wild-type (WT) and *Tet2^{Δ/Δ}p53^{Δ/Δ}* double knockout (DKO) mice, respectively. We purified Lineage^-Sca-1^+c-Kit^+ (LSK) hematopoietic stem/progenitor cells from WT and DKO mice 1 month after the tamoxifen injection and infected them with the *MYC-IRES-NGFR* and/or *RUNX2-IRES-GFP* retrovirus vectors (Fig. 6a). When we induced the differentiation of LSK cells into CD11b^-CD11c^+ B220^+ pDCs under liquid culture conditions with the Flt3-ligand and TPO (Fig. 6b)[10]. MYC plus RUNX2-expressing DKO cells produced more pDCs than the other cells following an additional treatment with lipopolysaccharide (LPS), which induced the terminal differentiation of pDCs (Fig. 6c). Furthermore, we found

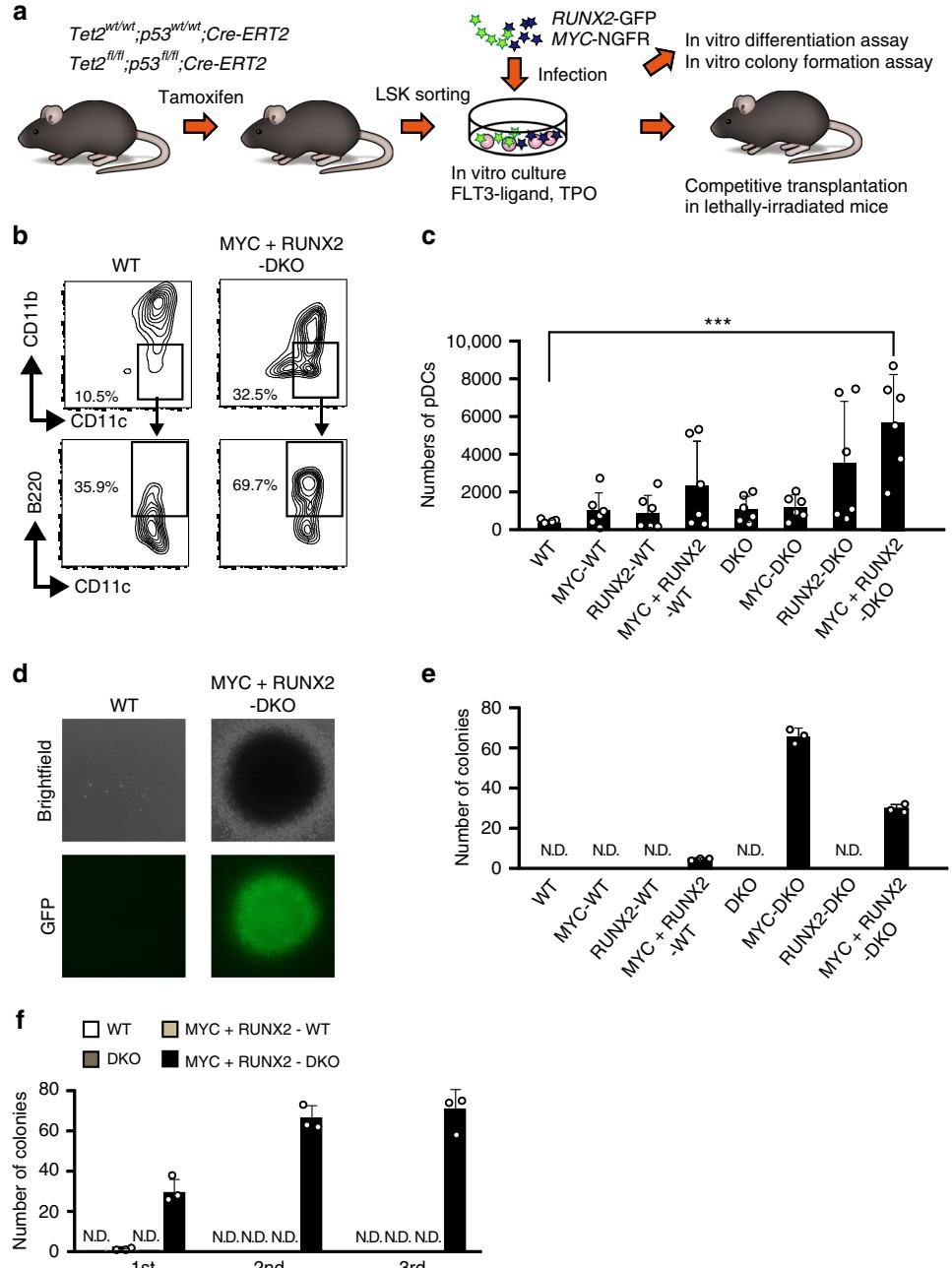

**Fig. 6** Transduction of MYC and RUNX2 promote the production of pDCs with the serial replating property. **a** Experimental scheme utilizing MYC and RUNX2 vectors and Tet2/p53 DKO mice for in vitro and in vivo assays. **b** Representative flow cytometric profiles of the CD45$^+$CD11b$^-$-gated CD11c$^+$B220$^+$ cells. **c** Absolute numbers of CD11b$^-$CD11c$^+$B220$^+$ pDCs among 1000 LSK cells-derived CD45$^+$ hematopoietic cells under liquid culture conditions ($n =$ 6). **d** Representative colony images (top columns) and GFP (bottom columns) of WT and DKO cells expressing MYC plus RUNX2 in the first plating. **e** Increased colony-forming properties of DKO cells expressing MYC and DKO cells expressing MYC plus RUNX2 ($n = 3$). **f** Enhanced serial re-plating properties of DKO cells expressing MYC plus RUNX2 (black bars) from those of WT cells (opened bars), WT cells expressing MYC plus RUNX2 (gray bars) and DKO (dark gray bars) ($n = 3$). **c**, **e**, **f** Bars show the mean±SD, *$p < 0.05$ and **$p < 0.01$ by the Student's $t$-test. N.D. not detected. Data are representative of 2–4 independent experiments

that *MYC*-expressing DKO cells and *MYC* plus *RUNX2*-expressing DKO cells showed higher numbers and larger colonies cultured with the Flt3-ligand and TPO (Fig. 6d, e). *MYC* plus *RUNX2*-expressing DKO cells subsequently gained serial re-plating capacities, whereas the other cells had severely diminished colonies (Fig. 6f). Thus, the overexpression of MYC and RUNX2 promoted the production of pDCs and the MYC overexpression was critical for the colony forming properties of DKO cells under this in vitro pDC differentiation setting.

**MYC and RUNX2 initiates the transformation of BPDCN in vivo.** We next examined the in vivo BPDCN-initiating capacity of MYC plus RUNX2 expressing DKO cells. After a 9-day culture promoting the differentiation of pDCs, we transplanted empty vector-transduced WT (WT), *MYC*- and *RUNX2*-transduced WT (MYC+RUNX2-WT), empty vector-transduced DKO (DKO), and *MYC*- and *RUNX2*-transduced DKO (MYC+RUNX2-DKO) CD45.2$^+$ cells into lethally irradiated recipient mice together with wild-type competitor CD45.1$^+$ bone marrow cells (Fig. 6a).

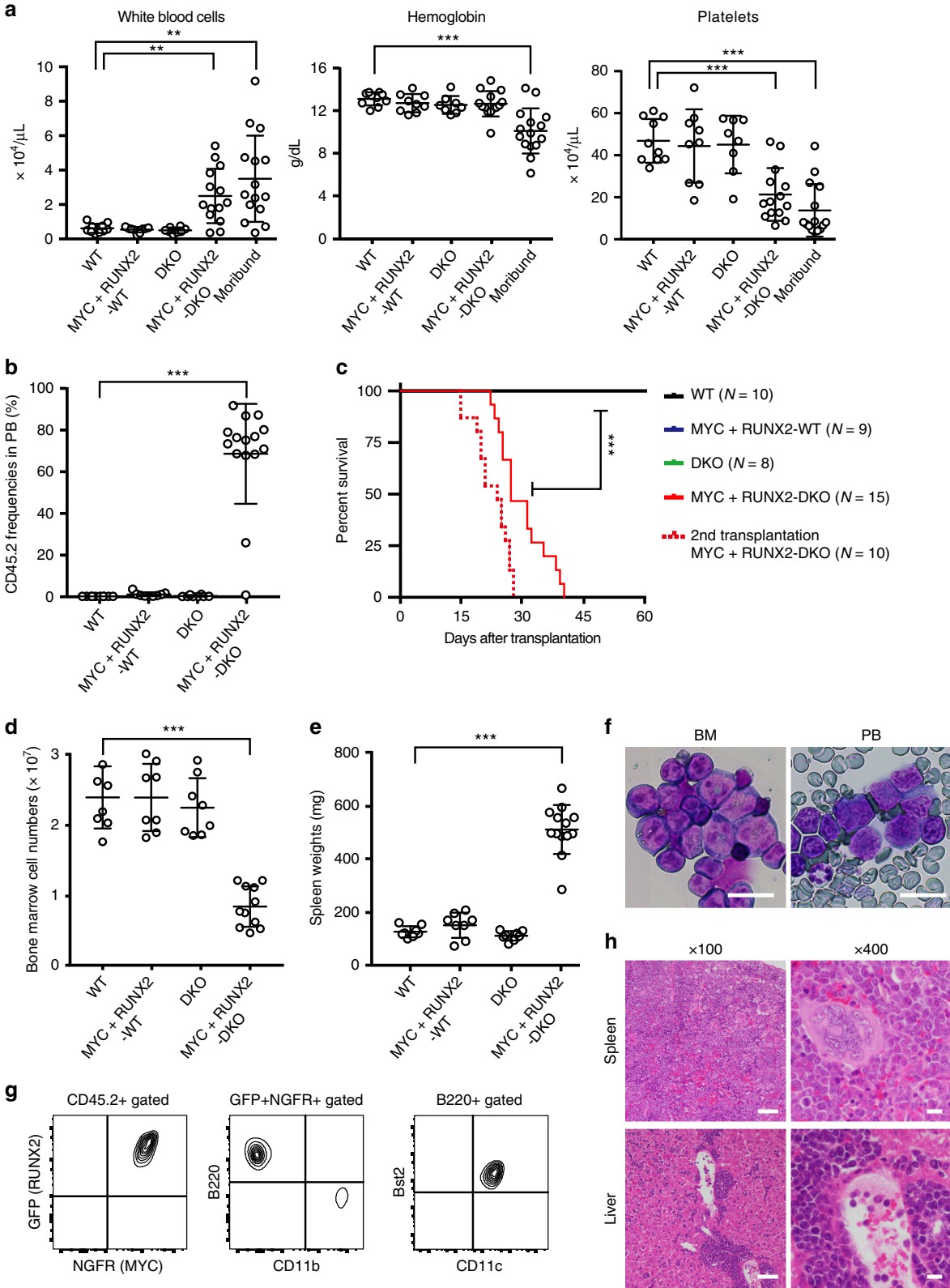

While WT, MYC+RUNX2-WT, and DKO mice did not show significant changes in blood counts in peripheral blood (PB) (Fig. 7a) due to the negligible populating capacities of CD45.2$^+$ cells 3 weeks post-transplantation (Fig. 7b), MYC+RUNX2-DKO moribund mice showed robust leukocytosis, anemia, and thrombocytopenia (Fig. 7a) following the emergence and expansion of CD45.2$^+$ cells (Fig. 7b). All MYC+RUNX2-DKO

mice died 1 month post-transplantation (Fig. 7c), whereas there were no lethal hematological malignancies in MYC+RUNX2-WT or DKO mice 6 months post-transplantation. At the time of killing, MYC + RUNX2-DKO moribund mice had significantly lower bone marrow cell counts and larger spleens than the other genotype mice (Fig. 7d, e). A detailed analysis of moribund mice revealed the expansion of immature leukemic blasts with a high

**Fig. 7** Transduction of MYC and RUNX2 promote the development of BPDCN in vivo. **a** Complete blood cell counts of WT ($n = 10$), MYC+RUNX2-WT ($n = 9$), DKO ($n = 8$), and MYC+RUNX2-DKO ($n = 15$) mice 3 weeks after transplantation and moribund MYC+RUNX2-DKO leukemic mice ($n = 15$) at the time of killing. **b** Proportions of CD45.2$^+$ cells in the PB of WT, MYC+RUNX2-WT, DKO, and MYC+RUNX2-DKO mice ($n = 8-10$) 3 weeks after transplantation and moribund MYC+RUNX2-DKO leukemic mice ($n = 15$) at the time of killing. **c** Significantly shorter median survival of primary-transplanted MYC-RUNX2-DKO mice ($n = 15$) (red line) than other genotype mice ($n = 8-10$) (25 days versus undetermined, ***$p < 0.0001$ by the Log-rank test) and a shorter median survival of the secondary-transplanted MYC-RUNX2-DKO mice ($n = 10$; broken red line). **d** Bone marrow cells counts isolated from 1 femur and 1 tibia of WT, MYC-RUNX2-WT, and DKO mice 6 months after transplantation ($n = 7-8$) and MYC-RUNX2-DKO leukemic mice at the time of killing ($n = 11$). **e** Spleen weights of WT, MYC-RUNX2-WT, and DKO mice 6 months after transplantation ($n = 7-8$) and MYC-RUNX2-DKO leukemic mice at the time of killing ($n = 13$). **f** Immature leukemic cells in the BM and the PB of MYC+RUNX2-DKO mouse observed by May-Grüenwald Giemsa staining. Scar bar, 20 μm. **g** Representative flow cytometry profiles of GFP$^+$NGFR-APC$^+$-gated Bst2$^+$B220$^+$CD11b$^-$CD11c$^+$ leukemic cells in a MYC+RUNX2-DKO mouse. **h** Representative histology of the spleen and liver of a MYC+RUNX2-DKO leukemic mouse observed by hematoxylin-eosin staining. Scar bars, 100 μm or 20 μm. **a**, **b**, **d**, **e** Bars show the mean±SD, *$p < 0.05$, **$p < 0.01$, and ***$p < 0.001$ by the Student's $t$-test. Data are combined from two independent experiments

---

nucleo-cytoplasmic ratio, dispersed chromatin, and microvacuoles in the PB and BM (Fig. 7f). A FACS analysis showed that these leukemic blasts were CD11b$^-$CD11c$^{mid/+}$B220$^+$Bst2$^+$ (Fig. 7g), which was consistent with the murine pDCs immunophenotype, and also massively infiltrated the spleen and liver tissues (Fig. 7h), whereas the leukemic cells expressed neither myeloperoxidase nor lysozyme; markers of myeloid cells (Supplementary Fig. 10). MYC+RUNX2-DKO leukemic cells were transplantable in secondary recipient mice with the same CD11b$^-$CD11c$^{mid/+}$B220$^+$Bst2$^+$ immunophenotype, and recipient mice died earlier at 2 weeks post-transplantation (the median survival, 23 days; Fig. 7c) due to severe anemia and thrombocytopenia (Supplementary Fig. 11). These results indicate that the overexpression of MYC and RUNX2 initiates the transformation of BPDCN in mice lacking Tet2 and p53.

**BPDCN activates oncogenes but impedes pDC-signature genes.** In order to elucidate the molecular mechanisms underlying the pathogenesis of MYC+RUNX2-DKO BPDCN, we performed an RNA sequencing analysis of CD11c$^{mid/+}$B220$^+$Bst2$^+$ leukemic cells isolated from distinct primary- and secondary-recipient mice in addition to those in LSK Flt3$^+$ lymphoid primed MPPs (LMPPs), macrophage-dendritic cell progenitors (MDPs), granulocyte–monocyte progenitors (GMPs), and pDCs isolated from WT mice. RNA sequencing from MYC+RUNX2-DKO leukemic cells revealed 1445 up-regulated genes and 1526 down-regulated genes from WT pDCs (Fig. 8a and Supplementary Data 2). In order to determine whether the gene expression profile of MYC+RUNX2-DKO leukemic cells reflected that in human BPDCN cells, we utilized published data sets comparing CAL-1 cells and normal pDCs in humans (Supplementary Data 2)[34], and found that up-regulated and down-regulated genes both significantly overlapped between these comparisons in patients and mice (Fig. 8a), confirming the relevance of this mouse model for human BPDCN.

A principal component analysis showed that primary and secondary BPDCN cells clustered together, and these BPDCN cells were located closer to WT LMPPs and MDPs but apart from GMPs (Fig. 8b). In addition, a hierarchical clustering analysis using differentially expressed genes between BPDCN cells and pDCs located BPDCN cells closer to WT LMPPs and MDPs than GMPs (Fig. 8c), supporting a distinct origin of BPDCN from the leukemic GMPs of AML[35]. Given the transduction of MYC in DKO leukemia cells, GSEA showed the positive enrichment of canonical MYC target genes in BPDCN cells from that in WT MDPs or WT pDCs (Fig. 8d and Supplementary Table 1), and a gene ontology analysis revealed significant enrichments in ribosome and translation-related pathways, which may be activated by MYC. Since human BPDCN cells showed lower expression levels of pDCs-signature genes than mature pDCs (e.g.

*TCF4*, *TLR7*, and *TLR9*; Supplementary Fig. 12), BPDCN cells showed the significant down-regulation of pDCs-signature genes expressed in WT pDCs, but the positive enrichment of these genes from those in LMPPs, MDPs, and GMPs (Fig. 8e and Supplementary Table 1). Unsupervised GSEA showed that interferon α and inflammatory responses were significantly and negatively enriched in BPDCN cells, which compromised the immunological function activated in mature pDCs (Fig. 8f and Supplementary Table 1). Quantitative RT-PCR confirmed that that the expression levels of pDC-signature genes, such as *Tcf4*, *Tlr7*, and *Tlr9*, were slightly increased in MYC+RUNX2-DKO leukemic cells and markedly increased in pDCs (Fig. 8g), indicating that BPDCN blocked terminal differentiation into pDCs by inhibiting the expression of pDC-signature genes to lower levels in mature pDCs. To examine the in vivo BPDCN-initiating capacity of precursor cells, we transplanted MDPs and pDCs purified from cultured MYC+RUNX2-DKO cells into lethally-irradiated recipient mice (Supplementary Fig. 13). While the pDCs did not repopulate in mice, the MDPs expanded in mice and developed lethal BPDCN with the same phenotype of CD11b$^-$CD11c$^{mid/+}$B220$^+$Bst2$^+$ following leukocytosis and thrombocytopenia (Supplementary Fig. 14). All MYC+RUNX2-DKO MDPs-transplanted mice died 70 days post-transplantation (median survival: 42 days versus undetermined in MYC+RUNX2-DKO pDCs mice) (Supplementary Fig. 15). Thus, MDPs were at least one of the cells of origin of BPDCN in this context.

The pDC-specific *RUNX2* super-enhancer appears to play an important role in the differentiation and function of normal pDCs via increasing the transcription of *RUNX2*, but in the leukemic cells harboring t(6;8), the *RUNX2* super-enhancer on the derivative 8 is hijacked to robustly activate the transcription of *MYC*, while the BENC enhancer does not appear to activate that of *MYC* (Fig. 8h). In addition, RUNX2 transcription factor binds to and helps the se*RUNX2*der8 activate the expression of *MYC* in BPDCN cells (Fig. 8h). Therefore, the expression of MYC and RUNX2 functions to confer survival and proliferative effects on leukemic precursors such as MDPs that initiated BPDCN in vivo, but also impedes terminal differentiation into pDCs to promote the development of BPDCN.

## Discussion

BPDCN is a rare and aggressive hematological malignancy, and shows a poor clinical outcome due to inappropriate chemotherapies because its pathogenesis has not yet been elucidated in detail. Since t(6;8)(p21;q24) is specific to BPDCN, using functional genomic assays combined with a mouse model, we elucidated the molecular mechanisms responsible for the initiation and progression of BPDCN, which are driven by the MYC oncogene and pDC-lineage transcription factor RUNX2.

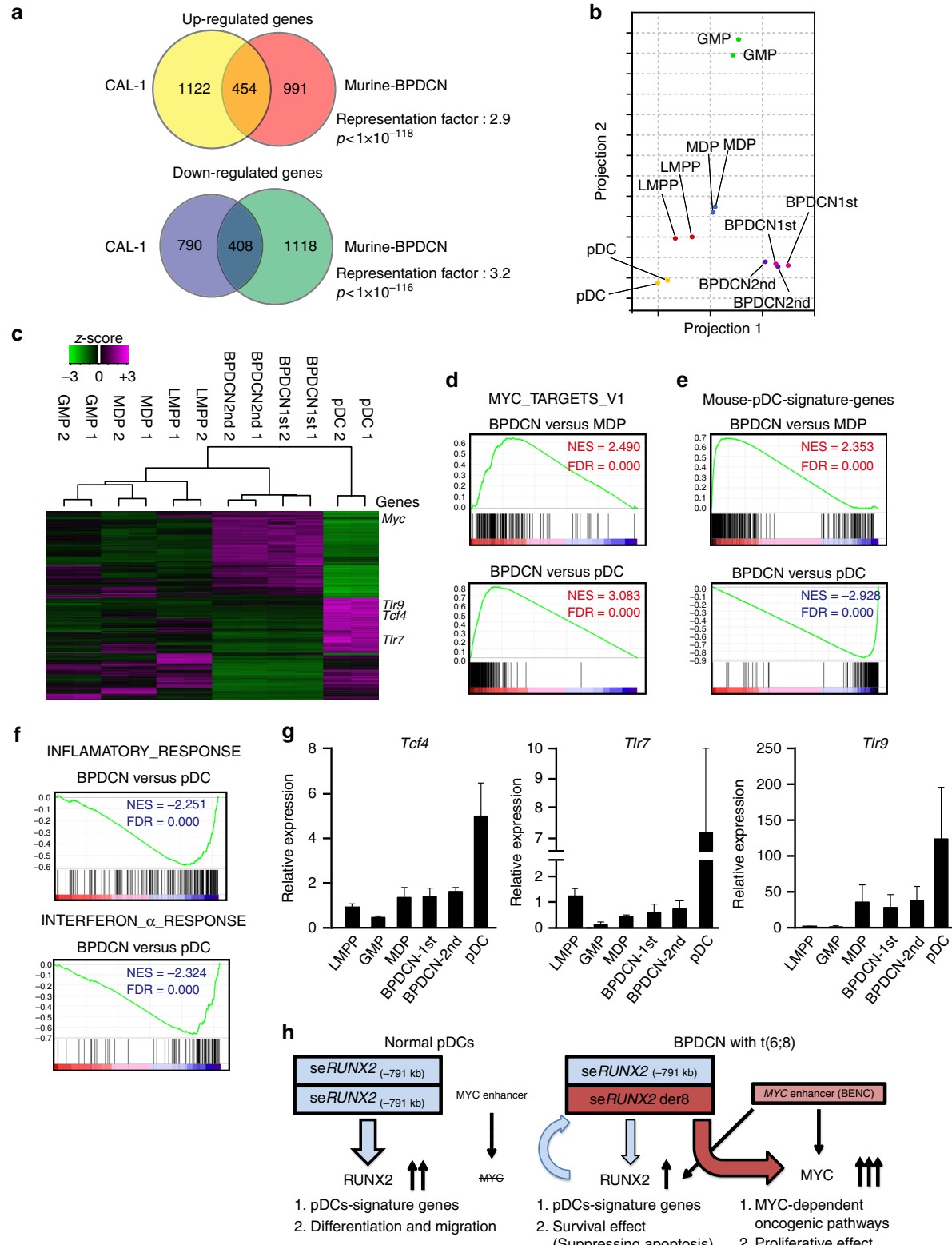

It has been recently reported that translocation (6;8)(p21;q24) was found in 22 out of 118 BPDCN patients and t(6;8)-positive BPDCN cells showed significantly increased expression of *MYC* compared to t(6;8)-negative BPDCN cells[8]. While translocation-induced enhancer hijacking has been reported in several tumors (e.g. IgH-*MYC* in lymphoma)[36], the mechanisms by which a translocation-induced enhancer activates a target oncogene and leads to transformation remain unclear. In the present study, we

demonstrated that t(6;8) caused the mutant-allele super-enhancer of *RUNX2* (se*RUNX2*der8) to activate the expression of *MYC* and promote the formation of BPDCN. While BPDCN cells did not show the Notch-dependent *MYC* enhancer observed in T-ALL[37], we found that CAL-1 BPDCN cells showed a H3K27ac peak of 1.7-Mb downstream enhancer of *MYC* equivalent to BENC module A enhancer, which was recently shown to mildly activate the expression of *Myc* in murine normal stem and progenitor

**Fig. 8** BPDCN cells activate MYC-mediated oncogenic pathways but impede the activation of pDC-signature genes. **a** Venn diagrams of significant overlaps in up-regulated genes and down-regulated genes in comparisons of MYC+RUNX2-DKO leukemic cells with murine pDCs and CAL-1 cells with human pDCs examined in GSE62014[34]. **b** A principal component analysis based on total gene expression in LMPPs (red dots), MDPs (blue dots), GMPs (green dots), and pDCs (yellow dots) isolated from wild-type mice and MYC+RUNX2-DKO leukemic cells isolated from two distinct primary mice (pink dots) and secondary transplanted mice (purple dots). **c** Hierarchical clustering based on differentially expressed genes between leukemic cells and wild-type pDCs (e.g. *Myc*, *Tcf4*, *Tlr7*, and *Tlr9*) in the same cells utilized in Fig. 8b. **d** GSEA plots for canonical MYC target genes comparing MYC+RUNX2-DKO leukemic cells to MDPs and pDCs isolated from WT mice. Normalized enrichment score (NES) and false discovery rate (FDR) q-values are indicated in Fig. 8d–f. **e** GSEA plots for pDC-signature genes, which were defined by the expression levels of genes in wild-type pDCs ($\geq$3-fold change, $p < 0.001$ significantly different from the levels of other fractions), comparing MYC+RUNX2-DKO leukemic cells to MDPs and pDCs isolated from WT mice. **f** GSEA plots for inflammatory response and interferon α response comparing MYC+RUNX2-DKO leukemic cells to WT pDCs. **g** Quantitative RT-PCR analysis of the expression of *Tcf4*, *Tlr7*, and *Tlr9* in LMPPs, MDPs, GMPs, and pDCs isolated from wild-type mice and MYC+RUNX2-DKO leukemic cells ($n = 3$). Bars show the mean±SD. **h** Model showing the functions of se*RUNX2* (−791 kb) in normal pDCs and se*RUNX2* der8 in BPDCN cells

cells[29], but also known as the Brd4-mediated enhancers of *Myc* in murine AML cells[30]. However, CAL-1 cells showed scattered BRD4 bindings in the BENC enhancers and did not change levels of BRD4 post the JQ1 treatment. Although the BENC enhancer may be able to increase the transcription of *MYC* on the WT allele and that of *RUNX2* on the mutant allele in BPDCN cells, we demonstrated that the specific deletion of the se*RUNX2*der8 significantly attenuated the transcription of *MYC* and resulted in the abolished cell growth capacities in vitro. Therefore, the *RUNX2* super-enhances appear to dominantly activate the transcription of both *MYC* and *RUNX2* in BPDCN cells (Fig. 8h).

Super-enhancers are critical for development and tissue homeostasis because they tightly regulate the expression of lineage-specific transcription factors in somatic stem cells, and the disruption of super enhancer regulation appears to contribute to malignant transformation[14]. The emergence and activation of the super-enhancer of *RUNX2* (−791 kb) appears to occur during the differentiation of pDCs with the subsequent induction of *RUNX2* expression, which is critical for the maturation and migratory function of pDCs[9,10]. Although BPDCN showed lower expression levels of *RUNX2* than pDCs, BPDCN cells had significantly higher expression levels of *RUNX2* than other subtypes of AML cells, and the knockdown of *RUNX2* significantly abrogated proliferative capacity following the attenuation of pDC-signature gene expression in BPDCN cells. In addition, the knockdown of *RUNX2* reduced the expression of the BPDCN-specific markers, CD56 and CD123, and inhibited the expression of canonical MYC target genes, indicating that RUNX2 executes the malignant transcriptional program in collaboration with MYC in BPDCN. *TCF4*, a master regulator of pDC differentiation, which is known to mutually regulate *RUNX2*, was recently identified as a critical transcription factor for the survival of BPDCN cells by RNAi screening[12]. We found that the knockdown of *RUNX2* reduced the expression of *TCF4* in BPDCN cells and this was accompanied by enhanced apoptosis. Thus, RUNX2-dependent transcriptional programs for the differentiation of pDCs were maintained in order to provide the survival properties of malignant cells, at least in part, due to the regulation of *TCF4*, but were also reprogramed to favor the propagation of BPDCN.

Since BPDCN cells have a higher sensitivity to the inhibition of BRD4, which has been shown to depend on TCF4[12], we also demonstrated that RUNX2 is one of the crucial targets in the setting of BRD4 inhibition in BPDCN cells. The inhibition of BRD4 significantly reduced the expression of *RUNX2* and *MYC* in BPDCN cells, and the ectopic expression of *RUNX2* alone, but not *MYC* alone, significantly restored impaired cell growth by the inhibition of BRD4, suggesting that MYC requires BRD4 and other factors to activate the enhancers and promote the cell growth, consistent with previous findings that ectopic expression of MYC did not restore JQ1-mediated impairments in cell growth

in BPDCN and prostate cancer cells[12,38]. We also found that the RUNX2-binding DNA motif was dominantly enriched in super-enhancers rather than in conventional enhancers in CAL-1 cells. Furthermore, RUNX2 itself bound to the region of the se*R-UNX2*der8, and the knockdown of *RUNX2* significantly reduced H3K27ac modification levels at the super-enhancer with subsequent reductions in *MYC*, suggesting a pioneering role for RUNX2 in the activation of super-enhancers during the transformation of BPDCN. Thus, RUNX2 plays a dominant role in controlling transcription networks in BPDCN, which provide the lineage-dependent survival signal and enhance the transformation capability driven by the expression of MYC.

MYC and RUNX transcription factor families both have the ability to activate the expression of p19[ARF], which stabilizes the p53 protein, and MYC is known to induce apoptosis in p53-dependent and independent manners. One decade ago, RUNX2 and MYC were shown to collaborate in order to generate T-cell lymphoma in a CD2 promoter-dependent manner in mice because RUNX2 suppressed MYC-dependent apoptosis in lymphoma cells in vivo[33]. Consistent with this finding, we demonstrated that the knockdown of *RUNX2* enhanced the levels of apoptosis in CAL-1 BPDCN cells harboring wild-type *TP53*, and the double knockdown of *RUNX2* and *MYC* markedly reduced the clonogenic capacity of CAL-1 cells. In addition, the transduction of RUNX2 and MYC transformed *p53* null/*Tet2* null cells, but not wild-type cells into BPDCN in vivo. These results imply that RUNX2 exerts a survival effect by suppressing MYC/p53-dependent apoptosis and, in turn, collaborates with MYC in this specific type of leukemia.

Since loss-of-function mutations in *TET2* and *TP53* are frequently detected in BPDCN patients, we revealed that the overexpression of *MYC* and *RUNX2* initiated the transformation of BPDCN-like disease lacking *Tet2* and *p53* in mice. Although we did not detect the expression of CD4, CD56 or CD123 in murine tumors, which are commonly detected in BPDCN patients, these leukemic cells showed the murine pDC phenotype of CD11b[−]CD11c[mid/+]B220[+]Bst2[+], accompanied by disseminative and transplantable capacities in secondary recipient mice. Interestingly, we observed that two out of 10 BPDCN mice showed focal invasions of leukemic cells on the skin and subcutaneous tissues. Since BPDCN-like tumor cells placed transcription profiles closer to LMPPs and MDPs than to pDCs and GMPs, we found that *MYC*- and *RUNX2*-expressing *p53* null/*Tet2* null MDPs repopulated in mice and developed lethal BPDCN disease, indicating leukemic MPDs being one of the cells of origin of BPDCN, which is also supported by a finding that *MYC*- and *RUNX2*-expressing *p53* null/*Tet2* null LSK cells and GMP cells occasionally developed AML in a non-pDC-differentiation setting. Therefore, this mouse model accurately recapitulates the phenotype of aggressive BPDCN and gives an insight into the molecular mechanisms underlying the pathogenesis of BPDCN.

A result of the present study is that translocation juxtaposes the pDCs-specific *RUNX2* super-enhancer with *MYC*, resulting in the development of BPDCN, and our mouse model recapitulates aggressive BPDCN lacking *p53* and *Tet2* in mice. We also demonstrated how MYC and RUNX2 collaborate for the initiation and progression of BPDCN by activating the function of super-enhancers that may be reversed through the inhibition of BRD4, as previously suggested[12,20]. These results support a rationale for combined applications of a BRD4 inhibitor with agents targeting MYC and/or RUNX2-dependent pDC-signature genes in patients (e.g. BCL2)[39].

## Methods

**Cells**. CAL-1, HEL, MOLM-13, MONO-MAC-1, and Jurkat cell lines were cultured in RPMI 1640 containing 10% fetal bovine serum (FBS) with penicillin/streptomycin in a humidified incubator. U2OS and Saos2 cell lines were cultured in DMEM containing 10% FBS. The CAL-1 cell line was established from a patient with BPDCN as previously described[6], and it is available from the author upon reasonable request. The CAL-1 was neither authenticated nor tested for mycoplasma contamination. The Jurkat was obtained from RIKEN (Japan), the other cell lines were obtained from ATCC.

**BPDCN patient samples**. All samples were obtained at Kumamoto University and Tokyo Medical University after written consent was obtained from patients. Patient anonymity was ensured, and the study was approved by the Institutional Review Committees at Kumamoto University (Kumamoto, Japan) and Tokyo Medical University (Tokyo, Japan).

**Virus vectors and infections**. MYC was constructed from cDNA encoding human *MYC*, and subcloned into the retrovirus vector pGCDNsam IRES-NGFR. RUNX2 was constructed from cDNA encoding human *RUNX2* and subcloned into the retrovirus vector pGCDNsam IRES-EGFP. shRNA directed against either *MYC* or *RUNX2* was expressed using the lentivirus vectors, pCS-H1-shRNA-EF-1-EGFP and pCS-H1-shRNA-EF-1-mRFP[40]. Target sequences were designated as follows: sh*RUNX2* #1; 5′-AAGGTTCAACGATCTGATTTG-3′, sh*RUNX2* #2; 5′-ACAA GGACAGAGTCAGATT-3′[41,42], sh*MYC* #1; 5′-CGATTCCTTCTAACAGA AATG-3′, sh*MYC* #2; 5′-GATGAGGAAGAAATCGATG-3′[12]. To produce the recombinant virus, virus vectors was transfected together with packaging vectors into 293T cells. Hematopoietic cells were then spinoculated with virus supernatant in 10 μg/ml protamine sulfate for 30 min, and infected cells were further incubated[43].

**Mice**. All mice were in the C57BL/6 background. Tp53 conditional KO (p53$^{flox/flox}$) mice were previously generated[44]. Tet2 conditional knockout (Tet2$^{flox/flox}$) mice were previously generated[45]. These mice were crossed with Rosa26:Cre-ERT2 mice (Taconic) for conditional deletion. C57BL/6 mice congenic for the Ly5 locus (CD45.1) were purchased from Sankyo-Lab Service. All experiments using these mice were performed in accordance with our institutional guidelines for the use of laboratory animals and approved by the Review Board for Animal Experiments of Kumamoto University (Kumamoto, Japan). All mice experiments were performed without randomization and blinding.

**pDCs differentiation and transplantation assays**. Purified murine hematopoietic stem/progenitor cells (Lineage⁻Sca-1⁺c⁻Kit⁺) were cultured in RPMI 1640 supplemented with 10% FBS, 50 μM 2-mercaptoethanol, 20 ng/ml human TPO, and 100 ng/ml Flt3-ligand for 9 days[10]. On day 3 of the in vitro culture, 3000 purified GFP⁺NGFR⁺ virus vector-infected MDP cells (Lineage⁻CD11c⁻c-Kit⁺CD115⁺CD135⁺) were injected into lethally irradiated CD45.1⁺ mice (8–12 weeks old) together with 2×10⁵ CD45.1⁺ bone marrow cells. On day 9 of the in vitro culture, 6000 purified GFP⁺NGFR⁺ virus vector-infected cells were injected into lethally irradiated CD45.1⁺ mice together with 2×10⁵ radioprotective CD45.1⁺ bone marrow cells. Colony formation assays were performed using Methocult M3234 (Stem Cell Technologies) supplemented with 20 ng/ml human TPO and 100 ng/ml Flt3-ligand with 1000 purified GFP⁺NGFR⁺c-Kit⁺ cells. In replating assays, colonies were counted on day 14 and 3×10⁴ pooled cells were plated into the same medium. Terminal pDC differentiation was induced by the administration of 1 ng/ml LPS for one day on day 9 of the culture with TPO and Flt3-ligand. In transplantation assays, 3000 purified GFP⁺NGFR⁺ virus vector-infected pDC cells (CD11b⁻CD11c⁺B220⁺) were injected into lethally irradiated CD45.1⁺ mice together with 2×10⁵ CD45.1⁺ bone marrow cells, and blood was analyzed using a Celtac-α (NIHON KODEN). Mice were excluded from the analysis if blood cells were not recovered 2 weeks post transplantation due to injection failure.

**Flow cytometry and immunohistochemistry antibodies**. Flow cytometry and cell sorting were performed by utilizing the following anti-human antibodies (clone and catalog numbers): CD271 (ME20.4, 345108), CD45 (HI30, 25-0459), CD4 (OKT4, 50-0048), CD56 (MY31, 60-0564), CD123 (6H6, 306011), and TCL1 (1-21, 330506), as well as the following anti-murine antibodies (clone and catalog numbers): CD45.2 (104, 109820), CD45.1 (A20, 110730), Gr1 (RB6-8C5, 108404), CD11b/Mac1 (M1/70, 101208), CD11c (N418, 117304), NK1.1 (PK136, 108704), Ter119 (116204), CD127/IL-7Rα (A7R34, 121104), Bst2 (927, 127010), B220 (RA3-6B2, 103212), CD3e (145-2C11, 100320), CD4 (L3T4, 100526), CD8α (53-6.7, 100714), CD56 (809220, FAB7820A), CD117/c-Kit (2B8, 105812), Sca1 (D7, 108114), CD34 (MEC14.7, 11-0341-85), CD123 (5B11, 106005), CD135 (A2F10, 135306), CD115 (AFS98, 135510), and FcγRII-III (93, 101308). These antibodies were purchased from BioLegend, eBioscience, R&D systems or TONBO biosciences. The lineage mixture solution contained biotin-conjugated anti-Gr1, B220, CD4, CD8α, Ter119, and IL-7Rα antibodies. Apoptotic cells were stained with an anti-Annexin V–APC antibody (BD 550474) and propidium iodide. In order to evaluate cell cycle progression, cells were cultured with BrdU, fixed and permeabilized according to the manufacturer's instructions, and then detected using the anti-BrdU-APC antibody (BD). All flow cytometric analyses and cell sorting were performed on FACSAriaII or FACSCantoII (BD 557892). Formalin fixed paraffin embedded bone marrow cells were stained with anti-MPO antibody (DAKO A0398) or anti-Lysozyme antibody (DAKO L1828).

**Chromatin IP sequencing**. CAL-1 cells were fixed by 0.5% paraformaldehyde at 37 °C for 2 min and then lysed. Cells were sonicated at 50% amplitude for 10 s 15 times. Samples were incubated with anti-RUNX2 antibody- (MBL; 8G5) or anti-H3K27Ac antibody-conjugated (Active motif; MABI 0309) Sheep anti-Rabbit IgG Dynabeads at 4 °C overnight. Input and immunoprecipitates were incubated at 65 °C for 4 h for reverse cross-linking, and DNA was purified using the MinElute PCR Purification Kit (Qiagen). ChIP-seq libraries were generated using the ThruPLEX DNA-seq kit (Rubicon Genomics). Bowtie2 (version 2.2.6;default parameters) was used to map the reads to the reference genome (UCSC/hg19). HOMER (version 4.9) was used for de novo motif discovery in the peaks of super enhancers and enhancers. ChIP-sequencing data were deposited in the DDBJ database under the accession number DRA006440 and DRA007469.

**Next-generation sequencing**. Next-generation sequencing was performed on CAL-1 cells using the Illumina HiSeq X Ten platform with a 150-bp paired-end read protocol according to the manufacturer's instructions. DNA-seq paired reads were aligned towards the UCSC hg19 reference genome using BWA-MEM. Gene fusions was detected by utilization of both soft clipping mapping and fragment information derived based on pair-end reads. Estimation of breakpoints for a gene fusion was performed after pileup of soft clipped reads for both sides. Genome sequencing data were deposited in the DDBJ database under the accession number DRA006440.

**RNA sequencing**. Total RNA was extracted using an RNeasy Plus Micro kit (Qiagen), and cDNA was synthesized using the SMARTer Pico PCR cDNA Synthesis kit (Clontech). ds-cDNA was fragmented and cDNA libraries were generated using the KAPA HyperPlus Library Preparation kit (KAPA Biosystems) and FastGene Adaptor kit (Fastgene). Sequencing was performed using Next-Seq500 (Illumina) with a single-read sequencing length of 60 bp, and the CLC genomic Workbench was used to analyze and visualize sequencing data. RNA sequencing data have been deposited in the DDBJ database under the accession number DRA006565.

**Microarray**. Twenty nanograms of total RNA was mixed with spike-in controls using the Agilent One Color Spike Mix Kit (Agilent), amplified, and labeled with Cyanine 3 using the Low Input Quick Amp Labeling Kit (Agilent) according to the manufacturer's instructions. A microarray analysis using the SurePrint G3 Human GE 8 × 60 K Microarray (Agilent) was performed according to the manufacturer's instructions. Microarray raw data were deposited in Gene Expression Omnibus under the accession number GSE110140.

**Chromosome conformation capture (3C)-quantitative PCR**. Cells were fixed by 1% formaldehyde at 25 °C for 10 min. Nuclei samples were digested with EcoRI at 37 °C 16 h and then ligated using T4 DNA ligase (NEB)[46]. Quantitative PCR primers were designed to detect the closest EcoRI target sites of the promoters and enhancers of *RUNX2* and *MYC* (Supplementary Table 2).

**CRISPR**. In order to delete the RUNX2 super enhancer from the translocated allele using the CRISPR-Cas9 system, the target sequence of gRNA was designed using sgRNA Designer[47] and CRISPR direct (Supplementary Table 2)[48]. Designed gRNAs were subcloned into the lentiCRISPRv2 vector[49], and transfected into 293T cells together with packaging plasmids. CAL-1 cells were infected using a sgRNA lentivirus and cultured in RPMI 1640 medium supplemented with 1 μg/ml puromycin in order to select sgRNA-vector-transduced clones. Uncropped gel images are available in Supplementary Fig. 16.

**FISH**. CAL-1 cells were treated with 0.05 μg/ml colcemid solution for 3 h and 75 mM KCl for 60 minutes, and were then fixed in 3:1 methanol:acetic acid solution. MYC and the enhancer region of MYC were detected using Vysis LSI-MYC dual color probes (Abbott Laboratories). The RUNX2 probe was generated by the bacterial artificial chromosome clone RP11-1019C24 (Empire Genomics) and labeled with DY-415-dUTP: aqua dye (Enzo Life Sciences). Probe-hybridized slides were washed, and cells were counterstained with 4′,6-diamidino-2-phenylindole (DAPI) and then analyzed using the BZ8000 System (Keyence).

**Quantitative RT-PCR and western blotting**. Total RNA was isolated using the RNeasy Mini kit and then reverse-transcribed by the ThermoScript RT-PCR system (Invitrogen) with an oligo-dT primer. Quantitative-RT-PCR was performed with LightCycler 480 (Roche) and SYBR Premix ExTaq (Takara). Expression levels were normalized to ACTB/β-actin expression. Primers for PCR are listed in Supplementary Table 2. The following antibodies were used for Western blotting: RUNX2 (MBL 8G5; 1:200 dilution), Actin (Santa Cruz C4; 1:1000 dilution), MYC (Santa Cruz 9E10; 1:100 dilution), H3 (Abcam ab1791; 1:500 dilution), and H3K27ac (Active motif MABI 0309; 1:1000 dilution). Uncropped immunoblot images are available in Supplementary Fig. 16.

**Statistical analysis**. All statistical tests were performed using Graph Pad Prism version 7 (GraphPad Software). The significance of differences was measured by an unpaired two-tailed Student's $t$-test or Mann–Whitney non-parametric test. A $P$ value of $<0.05$ was considered significant. No statistical methods were used to predetermine sample size for animal studies.

**Reporting Summary**. Further information on experimental design is available in the Nature Research Reporting Summary linked to this article.

## Data availability

All data are available within the article and Supplementary Information files. Sequencing data that support the findings of this study have been deposited in DDBJ or GEO under the accession numbers below. Whole-genome sequencing; DRA006440. Microarray; GSE110140. ChIP-sequencing; DRA006440 and DRA007469. RNA-sequencing; DRA006565

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

## Acknowledgements

We thank the members of the Sashida and Osato Laboratories for their discussions during the preparation of this manuscript and Ms. Mihoko Iimori, Ms. Noriko Yamamoto, Dr. Hirotaka Matsui, Mr. Shinji Kudoh, and Dr. Takaaki Ito for their technical help. This work was supported in part by a grant from the Takeda Science Foundation (S. K.), the Uehara Memorial Foundation (G.S.), Japanese Society of Hematology (M.O., G. S.), Grants-in-Aid for Scientific Research (16KT0113, 18H02842, 15H04312, 16H06276, 18KT0026, and 18K16090) and Strategic Research Foundation for Private Universities (S1511011) from the Ministry of Education, Culture, Sports, Science and Technology (MEXT) of Japan, and NCIS Centre Grant Seed Funding Program, Singapore Ministry of Education's AcRF Tier 3 grant (MOE2014-T3-1-006).

## Author contributions

S.K. performed experiments, analyzed results, and wrote the manuscript; K.T. performed experiments and provided reagents; T.U., T.YN. and Y.S. performed experiments; M.O., K.T.T., H.Y., A.K. and E.I. analyzed results; N.A., T.M. and N.N. provided reagents; A.I. analyzed results; K.O. provided reagents; M.O. designed the research and analyzed results; G.S. designed the research, performed experiments, analyzed results, and wrote the manuscript.

## Additional information

**Competing interests:** The authors declare no competing interests.

