## [Peer Review File · Nature Communications]

Reviewers' comments:

Reviewer #1, Expertise: Enhancers, leukemia, Myc (Remarks to the Author):

In this manuscript by Kubota and colleagues, the authors analyze the genetics of t(6;8)(p21;q24) translocation, which is a rare, but recurrent finding in BPDCN. The authors identify the breakpoints in patient samples and a cell line, and show that the RUNX2 enhancer aberrantly interacts with the MYC promoter by 3C experiments. They imply that the juxtaposed RUNX2 enhancer is amenable to therapy with BRD4 inhibitors, and show by CRISPR disruption of the mutant allele impairs colony formation. Lastly they show collaboration between RUNX2 and MYC in a p53/Tet2 DKO in vivo model of BPDCN. This rare disease is much understudied and this novel work makes a significant contribution to the field. On the whole, the experimental design is reasonable, though there are several areas where the data is over-interpreted and appropriate controls are missing. The paper is perhaps over-ambitious in its breadth and could have done with more detailed mechanistic analysis throughout, particularly on how MYC and RUNX2 potentially collaborate.

While the authors present data that the RUNX2 superenhancer activates MYC, the consequence of the translocation on RUNX2 expression is not really established. What role, if any, have the translocated MYC (BENC) enhancers got on RUNX2 expression? In Fig 1A, the authors show RUNX2 is less expressed in t(6;8) samples than in normal pDCs, suggesting the translocation may in fact lead to downregulation of RUNX2, in a similar mechanism through which GATA2 downregulation occurs as a consequence of EVI1 translocations in AML (see Groschel et al. Cell 2014).

Conceptually, it seems important to establish whether RUNX2 downregulation occurs and, and if so, whether this contributes to the disease phenotype. It may be that collaboration occurs with MYC at optimal RUNX2 levels, which are below physiological (essentially behaving like a haploinsufficient tumor suppressor), or that MYC overexpression in some way compensates for reduced RUNX2 expression. One potentially informative experiment would be to assess growth of CAL-1 cells when RUNX2 is expressed at the level of normal pDCs. Furthermore, if one were to assess the allelic ratios of heterozygous SNPs in RUNX2 mRNA from translocated samples, evidence of allelic skewing might help infer the transcriptional output resulting from enhancer swapping.

Further areas to address are outlined below:

Figure 1A: the wavy break in Y axis is unusual and need only be over the Y axis itself and the pDC bar. Relative expression is relative to what? It might be better shown as %GAPDH. Please clarify in the text why osteosarcoma cells are included as a comparison. It might be informative to look at MYC/RUNX2 expression in other BPDCN samples that do not have t(6;8) if any are available.

P6 end of paragraph 1: 'Furthermore, CAL-1 cells strongly expressed the RUNX2 and MYC proteins (Figure 1c), indicating that the expression of MYC and RUNX2 is increased during the development of BPDCN.' This is massively overstated and probably wrong: RUNX2 expression may be lower rather than increased during disease development compared to normal pDCs based on Fig 1A and impossible to say based on Fig 1C given lack of pDCs on the western blot. Same applies to P7: 'Based on the enhanced expression of MYC and RUNX2 in BPDCN cells harboring t(6;8),...'

Figure 1D: it would be helpful to have an illustration/cartoon of the entire derivative 6 and 8 chromosomes showing orientation of enhancers and genes in relation to centromere. Figure 1E – quality is quite poor for RUNX2. Fig 1F and G could potentially be moved to supplement.

Does RUNX2 bind to RUNX2 enhancer? Motif enrichment analysis of RUNX2 in Figure 4G suggests so.

Figure 1H. What are RUNX2 levels like in Jurkat cells? Is it expressed? Here I think it would be very helpful to address whether the MYC enhancer interacts with RUNX2 promoter.

A western blot for MYC is required (for 2a/b), as performed for RUNX2 (Fig 2d).

Given RUNX2 motif is overrepresented in super-enhancers in CAL-1 cells (Figure 4) – if RUNX2 is enriched at the RUNX2 SE, RUNX2 KD should have a reduction in MYC levels – have the authors looked at this? In which case, overexpression of MYC with a retrovirus might rescue RUNX2 KD. Equally it might be interesting to assess if RUNX2 overexpression can rescue MYC KD.

Is there synergy when both MYC and RUNX2 are knocked down? Otherwise the authors have no data to support the statement (P8) 'Therefore, MYC and RUNX2 collaborate to promote the development of BPDCN.'

Figure 3E: MYC signature is striking. Is MYC itself downregulated on RUNX2 KD?

Figure 4A: Reduction of MYC could be at the MYC-SE. It only suggests that these genes are regulated through enhancers, not necessarily the RUNX2-SE.

Figure 4F: what sort of protein levels are MYC and RUNX2 in the transduced cells? Could the author comment on the fact that RUNX2 overexpression has a better rescue than MYC overexpression?

Figure 5. What size deletions are present in each clone? Does RUNX2 expression change in these clones? Are there CRISPR-induced mutations at the sgRNA1 site on Chr 6? The reduction in MYC (5d) is not as marked as might be expected – can the authors comment on this? It is thus surprising that the colony output is so effected. Are there any heterozygous SNPs in MYC in this cell line that could help show allelic contribution to mRNA? It is not essential, but have the authors attempted to generate the translocation in a non t(6;8) DC line? These sort of experiments have been performed by others and can be very informative.

Figure 6: to show collaboration between MYC and RUNX2 surely we need to know what the contribution of each are individually? Same applies to Figure 7. Figure 6D – why is there no detectable GFP expression in WT- RUNX2+MYC cells?

Figure 8: P13 'In addition, a hierarchical clustering analysis using differentially expressed genes between BPDCN cells and pDCs located BPDCN cells closer to WT LMPPs and MDPs than GMPs (Figure 8c), supporting a distinct origin of BPDCN from the leukemic GMPs of AML.' I assume the authors mean leukemic LMPPs?

Reviewer #2, Expertise: AML, mouse models (Remarks to the Author):

In this study, the authors investigated the hypothesis that in blastic plasmacytoid dendritic cell neoplasm (BPDCN) with t(6;8), the fusion of a RUNX2 "super-enhancer" to regions near MYC drives MYC transcription. This hypothesis is in part based on the previous finding that RUNX2 expression is necessary for the development of plasmacytoid dendritic cells (pDC), the physiologic counterpart of BPDCN.

First, the authors identified a BPDCN cell line (CAL-1) that harbors t(6;8) along with two patient samples and confirmed the fusion of RUNX2 enhancer with Myc via FISH and WGS. Using CAL-1 as the model system, the authors demonstrate enhancer activation and looping to MYC promoter by 3C proximity ligation. Knocking down MYC and RUNX2 with shRNA decreased proliferation, and RUNX2 knockdown increased apoptosis. Transcriptome analysis of RUNX2 knockdown shows decreased expression levels of pDC gene signatures and MYC target gene signatures.

To test the function of the RUNX2 enhancer, the authors took two different approaches. First, JQ1 treatment at 250nM (intended to disrupt superenhancer activity) inhibited proliferation of CAL-1 but not Jurkat cells, and decreased MYC and RUNX2 transcription and protein levels. Reconstitution of RUNX2 expression appears to at least partially rescue cells from JQ1, while MYC has little effect on its own or in combination. The second approach demonstrates that deletion of RUNX2 superenhancer using CRISPR-CAS9 marked reduced MYC transcription.

Lastly, overexpression of RUNX2 and MYC in WT mouse hematopoietic cells, increases the replating ability of pDCs, and overexpression of MYC and RUNX2 in Tet2 and p53 knockout mice

(DKO) results in a fulminant leukemia that phenotypically resembles BPDCN. Interestingly, overexpression of MYC and RUNX2 in WT mice also results in an expansion of pDC like cells.

Comments:

This study can be divided into two distinct parts. The first part addresses the hijacking of a RUNX2 super-enhancer to drive MYC expression in cells with the t(6;8). The second part addresses the role of RUNX2 and MYC protein overexpression in the oncogenesis of BPDCNs. The authors provide robust evidence for the first part of the study. The second part of study is novel and interesting—in that BPDCN was induced with RUNX2 and MYC overexpression in the appropriate genetic background.

1. The results of the FISH analyses (Fig. 1d and S2) and the accompanying WGS data could be presented more clearly. The FISH data clearly document a break in MYC 3' of the MYC gene coding regions, which presumably leads to a translocation linking the RUNX2 associated supernenhancer to the MYC gene body (as supported by 3C data later). Inclusion of a cartoon showing the inferred structure of the der(6) and der(8) chromosomes would be helpful.
2. In figure 2A, the authors should document that MYC knockdown diminishes MYC protein level; this is the importance readout with respect to effects on function.
3. Typically, BAC clones are used as positive controls for 3C, but this is not possible in instances where abnormal chromosome rearrangements are present. However, the authors should as an additional confirmation sequence the products of the qPCR reactions to confirm that the products include the expected chromosome 6 and chromosome 8 regions.
4. There are some claims that are not supported by data shown. The authors claim that JQ1 treatment lowered H3K27ac in the RUNX2 enhancer; if true, these data should be shown. Also, the authors in the discussion state that RUNX2 binds to this enhancer; I see motif analysis, but I do not see any proof the RUNX2 binds to the enhancer.
5. The authors refer to the enhancer 1.7 Mb 3' of MYC as the BENC enhancer; is this the same as the BRD4-dependent enhancer described by Vakoc and colleagues in AML? Would be helpful to state so if true.
6. One of the most important findings in this study is the development of BPDCN-like leukemia in tet2 p53 DKO mice. There are some important markers that help to distinguish BPDCN from AML that are not described. It is important to show that the leukemias that emerge do not express myeloperoxidase or lysozyme. It would also be of interest to know if the murine tumors ever involve the skin, which is often seen in human BPDCN.
7. There is an inconsistency in figure 1f. The text stated hg19 reference genome was used rather than hg 18 as indicated by the figure.
8. There is a problem with language in some parts of the manuscript that sometimes make it difficult to grasp the authors' meaning. For example:
 - a. On page 8, "MYC and RUNX2 collaborate to promote the development of BPDCN"; this is a knockdown study in CAL1 cells; it tells you nothing about the development of BPDCN.
 - b. On page 9, "we investigated whether the BRD4 inhibitor JQ1 abrogated the formation of CAL-1 cells in vitro."
 - c. On page 14, "we show that the seRUNX2 (-791kb) aberrantly activates the transcription of RUNX2, which functions as a lineage-survival transcription factor." There is aberrant expression of MYC; it is unclear whether there is aberrant transcription of RUNX2.

Reviewer #1, Expertise: Enhancers, leukemia, Myc (Remarks to the Author):

In this manuscript by Kubota and colleagues, the authors analyze the genetics of t(6;8)(p21;q24) translocation, which is a rare, but recurrent finding in BPDCN. The authors identify the breakpoints in patient samples and a cell line, and show that the RUNX2 enhancer aberrantly interacts with the MYC promoter by 3C experiments. They imply that the juxtaposed RUNX2 enhancer is amenable to therapy with BRD4 inhibitors, and show by CRISPR disruption of the mutant allele impairs colony formation. Lastly they show collaboration between RUNX2 and MYC in a p53/Tet2 DKO in vivo model of BPDCN. This rare disease is much understudied and this novel work makes a significant contribution to the field. On the whole, the experimental design is reasonable, though there are several areas where the data is over-interpreted and appropriate controls are missing. The paper is perhaps over-ambitious in its breadth and could have done with more detailed mechanistic analysis throughout, particularly on how MYC and RUNX2 potentially collaborate. While the authors present data that the RUNX2 superenhancer activates MYC, the consequence of the translocation on RUNX2 expression is not really established. What role, if any, have the translocated MYC (BENC) enhancers got on RUNX2 expression? In Fig 1A, the authors show RUNX2 is less expressed in t(6;8) samples than in normal pDCs, suggesting the translocation may in fact lead to downregulation of RUNX2, in a similar mechanism through which GATA2 downregulation occurs as a consequence of EVI1 translocations in AML (see Groschel et al. Cell 2014). Conceptually, it seems important to establish whether RUNX2 downregulation occurs and, and if so, whether this contributes to the disease phenotype. It may be that collaboration occurs with MYC at optimal RUNX2 levels, which are below physiological (essentially behaving like a haploinsufficient tumor suppressor), or that MYC overexpression in some way compensates for reduced RUNX2 expression. One potentially informative experiment would be to assess growth of CAL-1 cells when RUNX2 is expressed at the level of normal pDCs. Furthermore, if one were to assess the allelic ratios of heterozygous SNPs in RUNX2 mRNA from translocated samples, evidence of allelic skewing might help infer the transcriptional output resulting from enhancer swapping.

Response: We appreciate the reviewer's comments on our work. We agree with the reviewer's comment on our misinterpretation of the dysregulation of RUNX2 expression in the pathogenesis of BPDCN in the original manuscript. The down-regulated expression of RUNX2 may contribute to the disease phenotype; however, we cannot conclude this point because we were unable to generate an inducible translocation in cells or in a murine model that develops BPDCN using a similar mechanism to EVI1/GATA2-mediated AML referred to by the reviewer. Therefore, we stated that the RUNX2 super-enhancer increased the expression of MYC via t(6;8), rather than the enhanced expression of MYC and RUNX2 promoting the formation of BPDCN, in the revised manuscript. Based on the other critical comments and suggestions

provided, we performed additional experiments and revised our manuscript as described below.

Further areas to address are outlined below:

Figure 1A: the wavy break in Y axis is unusual and need only be over the Y axis itself and the pDC bar. Relative expression is relative to what? It might be better shown as %GAPDH.

Response: We agree with the reviewer. We placed the break over the Y-axis and the normal pDC bar (Figure 1a). ACTB was used to normalize mRNA input for quantitative PCR and the expression levels of RUNX2 and MYC showed similar fold changes to those in HEL cells (Figures 1a and 1b). We described these points on page 6 and in the methodology on page 24.

Please clarify in the text why osteosarcoma cells are included as a comparison.

Response: We agree that this is an important point. U2OS and Saos2 cells have higher expression levels of RUNX2 than normal counterpart stromal cells, and Saos2 cells require RUNX2 to promote their cell growth capacity, as reported by Shin M, et al. PLoS Genetics 2016. We described this point for Figures 1a, 1b, and 1c on page 6.

It might be informative to look at MYC/RUNX2 expression in other BPDCN samples that do not have t(6;8) if any are available.

Response: We also agree that this is a very important point. A recent study published in *Leukemia* (Sakamoto K, et al. Leukemia 2018) reported that 22 out of 118 BPDCN patients harbored t(6;8) and had higher expression levels of MYC than BPDCN cells without t(6;8). We discussed this novel finding on page 6 and added this study as a reference.

P6 end of paragraph 1: 'Furthermore, CAL-1 cells strongly expressed the RUNX2 and MYC proteins (Figure 1c), indicating that the expression of MYC and RUNX2 is increased during the development of BPDCN.' This is massively overstated and probably wrong: RUNX2 expression may be lower rather than increased during disease development compared to normal pDCs based on Fig 1A and impossible to say based on Fig 1C given lack of pDCs on the western blot. Same applies to P7: 'Based on the enhanced expression of MYC and RUNX2 in BPDCN cells harboring t(6;8),...'

Response: We fully agree with the reviewer and apologize for our overstatement in the original manuscript. As the reviewer pointed out, RUNX2 expression levels in BPDCN cells were higher than myeloid progenitors and precursors of pDCs, but lower than normal counterpart pDCs. Therefore, we stated that the expression of MYC increased with the development of BPDCN cells and described this point in the results on pages 6 and 7 in the revised manuscript.

Figure 1D: it would be helpful to have an illustration/cartoon of the entire derivative 6 and 8 chromosomes showing orientation of enhancers and genes in relation to centromere.

Response: We appreciate the reviewer's suggestion. We added an illustration of derivatives 6 and 8 showing the orientations of enhancers and RUNX2 and MYC to Figure 1d.

Figure 1E – quality is quite poor for RUNX2.

Response: We agree that the FISH image did not clearly show RUNX2 signals. We added another FISH image showing better RUNX2 signals in interphase from the same patient to Figure 1e, and moved the original image to Supplementary Figure S2.

Fig 1F and G could potentially be moved to supplement.

Response: We agree with the reviewer. We moved the murine H3K27ac ChIP sequencing image to Supplementary Figure S5 in the revised manuscript.

Does RUNX2 bind to RUNX2 enhancer? Motif enrichment analysis of RUNX2 in Figure4G suggests so.

Response: We appreciate the reviewer's comment, which was also raised by the other reviewers. We performed ChIP sequencing on CAL-1 BPDCN cells using an anti-RUNX2 antibody and found that RUNX2 was significantly enriched within the region of the RUNX2 super-enhancer showing at least 5 peaks. We described this point on page 10, added RUNX2 and H3K27ac ChIP sequencing images in the RUNX2 super-enhancer in CAL-1 cells to Figure 4f, and moved motif enrichment analysis data to Supplementary Figure S8.

Figure 1H. What are RUNX2 levels like in Jurkat cells? Is it expressed? Here I think it would be very helpful to address whether the MYC enhancer interacts with RUNX2 promoter.

Response: We agree that this is an important point. As shown in Figure 4b, Jurkat cells showed markedly lower/undetectable RUNX2 protein levels than CAL-1 cells; therefore, the amplified product of 3C-q-PCR in Jurkat cells in Figure 1h, left (now it is Figure 1g) does not appear to be functional for activating the expression of RUNX2, but is rather due to a non-specific association between these cis regions on chromosome 6 after a ligase treatment in Jurkat cells. We also found that CAL-1 BPDCN cells showed a single peak of H3K27ac in the BENC MYC enhancer that lacked significant BRD4 enrichment, as shown in Supplementary Figure S4 in the revised manuscript. We attempted to examine the relationship between the BENC MYC enhancer and RUNX2 promoter, but did not observe any significant amplification of 3C-q-PCR in CAL-1 cells, suggesting that the BENC enhancer activated the expression of RUNX2 in CAL-1 cells to a limited degree, even if it still associated with the promoter of RUNX2. We described these points in the results on page 7 and in the discussion on pages 16 to 17.

A western blot for MYC is required (for 2a/b), as performed for RUNX2 (Fig 2d).

Response: We agree with the reviewer. We performed a Western blot analysis and added Western blot images for MYC in shRNA-MYC-transduced cells to Figure 2b.

Given RUNX2 motif is overrepresented in super-enhancers in CAL-1 cells (Figure4) – if RUNX2 is enriched at the RUNX2 SE, RUNX2 KD should have a reduction in MYC levels – have the authors looked at this?

Response: We agree with the reviewer. Consistent with the enrichment of RUNX2 in the super-enhancer of RUNX2 shown in Figure 4f, RUNX2-KD decreased the expression levels of MYC in CAL-1 BPDCN cells, as shown in Figure 3f in the revised manuscript.

In which case, overexpression of MYC with a retrovirus might rescue RUNX2 KD. Equally it might be interesting to assess if RUNX2 overexpression can rescue MYC KD.

Response: We agree that this is an important and interesting point. We transduced MYC in RUNX2-KD CAL-1 BPDCN cells and examined the colony formation capacity of these cells *in vitro*. The overexpression of MYC partially rescued the reduced colony formation capacity of RUNX2-KD cells, as shown here (**Right figure**). We added this information to the results on page 9.

Is there synergy when both MYC and RUNX2 are knocked down? Otherwise the authors have no data to support the statement (P8) 'Therefore, MYC and RUNX2 collaborate to promote the development of BPDCN.'

Response: We fully agree with the reviewer and apologize for our overstatement in the original manuscript. We performed the knockdown of RUNX2 and MYC expression in CAL-1 cells, and found that RUNX2/MYC-KD cells showed a significantly weaker colony formation capacity than single KD cells, as the reviewer predicted. We described this point on pages 7 to 8 and added this result to Figure 2j.

Figure 3E: MYC signature is striking. Is MYC itself downregulated on RUNX2 KD?

Response: Since RUNX2-KD cells showed significant reductions in MYC signature genes, RUNX2 KD decreased MYC mRNA expression levels in CAL-1 cells. We added this quantitative PCR data on MYC to page 9 and Figure 3f.

Figure 4A: Reduction of MYC could be at the MYC-SE. It only suggests that these genes are regulated through enhancers, not necessarily the RUNX2-SE.

Response: We appreciate the reviewer's comment. While we and another group demonstrated that CAL-1 cells showed a relatively small enhancement in BENC, the BRD4 peaks of which

were not sensitive to the JQ1 treatment in Supplementary Figure S4 (reported by Dr. Staudt's group in Ceribelli, Cancer Cell 2016), we agree with the reviewer that the expression of MYC was not necessarily regulated by the RUNX2 super-enhancer based on reduced, but sustained expression levels following the JQ1 treatment in Figure 4a. We corrected our statement and described the results obtained on pages 9 and 10 and in the discussion on pages 16 and 17.

Figure 4F: what sort of protein levels are MYC and RUNX2 in the transduced cells? Could the author comment on the fact that RUNX2 overexpression has a better rescue than MYC overexpression?

Response: We agree that these are important points. We examined RUNX2 and MYC mRNA and protein levels in transduced cells, showing single-transduced cells here (**Right figure**; bars graphs showing q-PCR for MYC and RUNX2) and double-transduced cells in Figure 4f (now Figure 4g).

The efficacies of the transduction of RUNX2 and MYC were similar based on their protein levels in transduced cells. Furthermore, a very interesting observation was that the overexpression of RUNX2

restored the impaired growth of CAL-1 cells more effectively than that of MYC. We consider RUNX2 to have a potentially potent function in activating enhancers and ensuring the survival of this specific cancer cell; however, the underlying molecular mechanisms by which RUNX2 restores the survival and growth of BPDCN cells upon the inhibition of BRD4 have not yet been elucidated. Other groups also reported similar findings showing that MYC did not restore impairments in cell growth mediated by JQ in prostate cancer and BPDCN using the same CAL-1 cell line (Asangani, et al. Nature 2014; Ceribelli, et al. Cancer Cell 2016). We described these points in the results on page 10 and in the discussion on page 17 and 18.

Figure 5. What size deletions are present in each clone? Does RUNX2 expression change in these clones? Are there CRISPR-induced mutations at the sgRNA1 site on Chr 6? The reduction in MYC (5d) is not as marked as might be expected – can the authors comment on this? It is thus surprising that the colony output is so effected.

Response: We thank the reviewer for these comments. We added the deletion sizes of these clones to Figure 5c, which were the expected sizes based on a combination of sgRNA vectors. We also confirmed the expected 18-bp deletion of the sgRNA #1 site on chromosome 6 by performing Sanger sequencing

for that region, as shown here (**Right figure**). By utilizing

sgRNA#1-induced cells, we demonstrated that single sgRNA #1-transduced cells did not change RUNX2 expression levels and grew as well as control cells, as shown here (**Left figure**). Furthermore, since the reviewer pointed out that colony formation was markedly inhibited as expected by the reduced levels of MYC, we noted that these clones showed between 20 and 60% reductions in RUNX2 expression levels, as shown here (**Right figure**), suggesting that

MYC positively regulates the expression of RUNX2 in BPDCN cells in a direct and/or an indirect manner. Therefore, the reduced expression of MYC and RUNX2 may further inhibit colony formation by RUNX2

super-enhancer-deleted clones under this experimental setting; however, the precise molecular mechanisms responsible for the MYC-dependent regulation of RUNX2 in BPDCN remain unclear. We described these points on page 11.

Are there any heterozygous SNPs in MYC in this cell line that could help show allelic contribution to mRNA?

Response: We agree that this is an important point. We analyzed SNPs in MYC in the CAL-1 BPDCN cell line using all of our genome sequencing data, but did not find heterozygous SNPs of MYC in CAL-1.

It is not essential, but have the authors attempted to generate the translocation in a non t(6;8) DC line? These sort of experiments have been performed by others and can be very informative.

Response: We agree that this is a very interesting point. However, due to the limited time available for revisions, we were unable to generate and examine a mutant cell line that induces the translocation (6;8). We intend to generate this model in future studies in order to obtain a clearer understanding of the function and transcriptional regulation of MYC and RUNX2 during the differentiation of normal pDCs as well as the development of BPDCN.

Figure 6: to show collaboration between MYC and RUNX2 surely we need to know what the contribution of each are individually? Same applies to Figure 7. Figure 6D – why is there no detectable GFP expression in WT- RUNX2+MYC cells?

Response: We fully agree with the reviewer. In addition to the double transduction, we performed the individual transduction of RUNX2 and MYC into WT and DKO cells, and found that MYC-expressing DKO cells and MYC plus RUNX2-expressing DKO cells showed greater numbers of and larger colonies cultured with the Flt3-ligand and TPO in Figures 6d and 6e.

Thus, the overexpression of MYC and RUNX2 promoted the production of pDCs and MYC overexpression was critical for the colony-forming properties of DKO cells under this *in vitro* pDC differentiation setting. We described these results on page 12. Regarding the same issue for Figure 7, we also examined the *in vivo* disease-initiating capacity of MYC-expressing DKO cells, and found that the majority of these cells developed lethal BPDCN disease in mice (n=10), similar to MYC plus RUNX2-expressing DKO cells in mice. Although these results are very interesting, this experiment is still ongoing; therefore, we did not include data on single transduction in the revised manuscript. Regarding the image in Figure 6D, we apologize for its low quality because the MYC-RUNX2-WT colony showed scant/undetectable GFP expression due to smaller numbers of cells in the colony. We deleted this image from the revised version.

Figure 8: P13 'In addition, a hierarchical clustering analysis using differentially expressed genes between BPDCN cells and pDCs located BPDCN cells closer to WT LMPPs and MDPs than GMPs (Figure 8c), supporting a distinct origin of BPDCN from the leukemic GMPs of AML.' I assume the authors mean leukemic LMPPs?

Response: We thank the reviewer for this comment on leukemia-initiating cells. Due to the limited number of LMPPs in that *in vitro* transduction and differentiation setting, we purified MYC and RUNX2-expressing DKO MDPs and pDCs (3000 cells each) and injected in lethally irradiated mice together with 2×10^5 wild-type bone marrow cells, as shown in Supplementary Figure S12, and found that all MDP-transplanted mice developed lethal BPDCN disease with a short latency, whereas pDCs did not repopulate in mice or result in the development of BPDCN, as shown in Supplementary Figures S13 and S14 (performed in two independent experiments). Consistent with the gene expression profiles of BPDCN cells being closer to MDPs and LMPPs, MDPs were at least one of the cells of origin of BPDCN in this context. We described these points in the results on page 14 and 15 and in the discussion on page 19.

Reviewer #2, Expertise: AML, mouse models (Remarks to the Author):

In this study, the authors investigated the hypothesis that in blastic plasmacytoid dendritic cell neoplasm (BPDCN) with t(6;8), the fusion of a RUNX2 “super-enhancer” to regions near MYC drives MYC transcription. This hypothesis is in part based on the previous finding that RUNX2 expression is necessary for the development of plasmacytoid dendritic cells (pDC), the physiologic counterpart of BPDCN.

First, the authors identified a BPDCN cell line (CAL-1) that harbors t(6;8) along with two patient samples and confirmed the fusion of RUNX2 enhancer with Myc via FISH and WGS. Using CAL-1 as the model system, the authors demonstrate enhancer activation and looping to MYC promoter by 3C proximity ligation. Knocking down MYC and RUNX2 with shRNA decreased proliferation, and RUNX2 knockdown increased apoptosis. Transcriptome analysis of RUNX2 knockdown shows decreased expression levels of pDC gene signatures and MYC target gene signatures.

To test the function of the RUNX2 enhancer, the authors took two different approaches. First, JQ1 treatment at 250nM (intended to disrupt superenhancer activity) inhibited proliferation of CAL-1 but not Jurkat cells, and decreased MYC and RUNX2 transcription and protein levels. Reconstitution of RUNX2 expression appears to at least partially rescue cells from JQ1, while MYC has little effect on its own or in combination. The second approach demonstrates that deletion of RUNX2 superenhancer using CRISPR-CAS9 marked reduced MYC transcription.

Lastly, overexpression of RUNX2 and MYC in WT mouse hematopoietic cells, increases the replating ability of pDCs, and overexpression of MYC and RUNX2 in Tet2 and p53 knockout mice (DKO) results in a fulminant leukemia that phenotypically resembles BPDCN. Interestingly, overexpression of MYC and RUNX2 in WT mice also results in an expansion of pDC like cells.

Response: We appreciate the reviewer’s comments on our work, according to which we performed additional experiments and revised our manuscript as described below.

Comments:

This study can be divided into two distinct parts. The first part addresses the hijacking of a RUNX2 super-enhancer to drive MYC expression in cells with the t(6;8). The second part addresses the role of RUNX2 and MYC protein overexpression in the oncogenesis of BPDCNs. The authors provide robust evidence for the first part of the study. The second part of study is novel and interesting—in that BPDCN was induced with RUNX2 and MYC overexpression in the appropriate genetic background.

1. The results of the FISH analyses (Fig. 1d and S2) and the accompanying WGS data could be presented more clearly. The FISH data clearly document a break in MYC 3’ of the MYC gene

coding regions, which presumably leads to a translocation linking the RUNX2 associated superenhancer to the MYC gene body (as supported by 3C data later). Inclusion of a cartoon showing the inferred structure of the der(6) and der(8) chromosomes would be helpful.

Response: We agree with the reviewer. Since Reviewer #1 also mentioned this point, we added an illustration of derivatives 6 and 8 showing the orientations of the enhancers and RUNX2 and MYC to Figure 1d in order to help readers understand the relationship between the MYC gene and RUNX2 super-enhancer in BPDCN cells with t(6;8).

2. In figure 2A, the authors should document that MYC knockdown diminishes MYC protein level; this is the importance readout with respect to effects on function.

Response: We fully agree with the reviewer. We added a Western blot of MYC protein expression levels in shRNA-MYC-transduced CAL-1 cells to Figure 2b.

3. Typically, BAC clones are used as positive controls for 3C, but this is not possible in instances where abnormal chromosome rearrangements are present. However, the authors should as an additional confirmation sequence the products of the qPCR reactions to confirm that the products include the expected chromosome 6 and chromosome 8 regions.

Response: We agree with the reviewer. We were unable to utilize BAC clones, and instead performed 3C followed by target sequencing to detect the expected chromosome 6 and 8 regions using CAL-1 cells. The results obtained demonstrated that the RUNX2 super-enhancer physically associated with the MYC promoter on der(8) in CAL-1 cells, but not in Jurkat cells, as shown in Figure 1g and Supplementary Figure S6. We described this point on page 7.

4. There are some claims that are not supported by data shown. The authors claim that JQ1 treatment lowered H3K27ac in the RUNX2 enhancer; if true, these data should be shown. Also, the authors in the discussion state that RUNX2 binds to this enhancer; I see motif analysis, but I do not see any proof the RUNX2 binds to the enhancer.

Response: We fully agree with the reviewer. The JQ1 treatment reduced H3K27ac expression levels, as shown in Figure 4e. We also performed H3K27ac ChIP sequencing on CAL-1 cells with or without the JQ1 treatment, and found that the JQ1 treatment significantly reduced the levels of H3K27ac and BRD4, as previously examined by Ceribelli M, et al. Cancer Cell 2016 in the RUNX2 super-enhancer in CAL-1 cells. We added these results to Figure 4f and described this point on page 10. We then performed RUNX2 ChIP sequencing on CAL-1 cells and found that at least 5 peaks of the RUNX2 enrichment in the RUNX2 super-enhancer, as shown in Figure 4f, as RUNX2-binding regions were more strongly enriched in super-enhancers than in normal enhancers, as shown in Supplementary Figure S7, which is consistent with an observation in the motif-enrichment analysis of H3K27ac peaks, which was moved to Supplementary Figure S8 in the revised version. We described these points on page 10.

5. The authors refer to the enhancer 1.7 Mb 3' of MYC as the BENC enhancer; is this the same as the BRD4-dependent enhancer described by Vakoc and colleagues in AML? Would be helpful to state so if true.

Response: We thank the reviewer for this comment. We downloaded and mapped their data on mm9, and confirmed that Vakoc's clustered enhancers of Myc in AML were located on the same regions of the BENC enhancer (containing A, B, C, D, F, and G modules) described by Trumpp's group, as shown in Supplementary Figure S4 (lower panel). We stated this point in the results on page 6 and 7 and in the discussion on page 16.

6. One of the most important findings in this study is the development of BPDCN-like leukemia in tet2 p53 DKO mice. There are some important markers that help to distinguish BPDCN from AML that are not described. It is important to show that the leukemias that emerge do not express myeloperoxidase or lysozyme. It would also be of interest to know if the murine tumors ever involve the skin, which is often seen in human BPDCN.

Response: We thank the reviewer for these comments. We performed the staining of myeloperoxidase in DKO BPDCN and wild-type normal bone marrow tissues and found that these leukemic cells did not express myeloperoxidase, as shown in Supplementary Figure S9, as the reviewer predicted. We also attempted to establish whether these murine leukemia cells infiltrate skin. Although two out of 10 mice showed focal invasions of leukemic cells frequently showing mitosis on the skin and subcutaneous tissues (**Bottom pictures**), we described this observation in the discussion on page 19 because we only completed the examination of one cohort during the revision period. Nevertheless, these results further support our murine model reflecting human BPDCN.

7. There is an inconsistency in figure 1f. The text stated hg19 reference genome was used rather than hg 18 as indicated by the figure.

Response: We agree with the reviewer. We corrected this inconsistency and used hg19 in Figure 1f.

8. There is a problem with language in some parts of the manuscript that sometimes make it difficult to grasp the authors' meaning. For example:

a. On page 8, "MYC and RUNX2 collaborate to promote the development of BPDCN"; this is a knockdown study in CAL1 cells; it tells you nothing about the development of BPDCN.

Response: We apologize for our unclear meanings in the manuscript. We agree with the reviewer regarding this point, and corrected it to "cell growth" or "formation" on pages 7 and 8 in the revised version.

b. On page 9, "we investigated whether the BRD4 inhibitor JQ1 abrogated the formation of CAL-1 cells in vitro."

Response: We corrected this sentence to "we examined whether the BRD4 inhibitor JQ1 abrogated the formation of CAL-1 cells." on page 9.

c. On page 14, "we show that the seRUNX2 (-791kb) aberrantly activates the transcription of RUNX2, which functions as a lineage-survival transcription factor." There is aberrant expression of MYC; it is unclear whether there is aberrant transcription of RUNX2.

Response: We fully agree with the reviewer. We corrected it to "the *RUNX2* super-enhancer on derivative 8 is hijacked to robustly activate the transcription of *MYC*, while the BENC enhancer does not appear to activate that of *MYC*." on page 15 of the revised manuscript.

Reviewers' comments:

Reviewer #1 (Remarks to the Author):

The authors have significantly improved the manuscript and toned down their previous interpretation of the data. There are still a few minor improvements required:

All ChIP-seq tracks need a numbered Y-axis

Page 7 - reference to FIG 1g and FIG S6 do not match what is referred to in the text

Fig S3 - typos on X axis 'Ditected enhacer in enhancer'

Page 9- 'abrogated the formation of CAL1-cells' : do the authors mean 'abrogated the proliferation of CAL-1 cells'

Page 7- 'inherit the super enhancer'. Do not think inherit is the correct term here

Reviewer #2 (Remarks to the Author):

In the revised manuscript, the authors have been largely responsive to the points and concerns raised in my original critique. Ideally, I would like to see them do some additional characterization of the immunophenotype of the tumors that arise in the mouse model. Other highly characteristic features of human BPDCN is expression of CD4, CD56, CD123, and TCL1, and failure to express lysozyme; some of these can be rapidly tested by flow cytometry and/or IHC.

Reviewer #1 (Remarks to the Author):

The authors have significantly improved the manuscript and toned down their previous interpretation of the data. There are still a few minor improvements required:

Response: We appreciate the reviewer's comments on our revised manuscript. We corrected our manuscript as described below.

All ChIP-seq tracks need a numbered Y-axis

Response: We agree with the reviewer. We added numbers on Y-axis in all ChIP-sequencing tracks in Figure 1, Figure 4, Supplementary Figure 4, and Supplementary Figure 5.

Page 7 - reference to FIG 1g and FIG S6 do not match what is referred to in the text

Response: We thank the reviewer for this comment. We moved that reference to Figure 1g and Figure S6 and added a new reference to Figure 4b on Page 7.

Fig S3 - typos on X axis 'Ditected enhacer in enhancer'

Response: Response: We thank the reviewer for this comment. We corrected to "Rank of detected enhancers" on the X-axis in Figure S3.

Page 9- 'abrogated the formation of CAL1-cells' : do the authors mean 'abrogated the proliferation of CAL-1 cells'

Response: We agree with the reviewer regarding to this point. We changed it to that JQ1 abrogated the proliferation of CAL-1 cells on Page 9 and Page 10.

Page 7- 'inherit the super enhancer'. Do not think inherit is the correct term here

Response: We agree with the reviewer that inherit is not a proper term here. We changed to state that 'BPDCN cells utilize the super enhancer of *RUNX2* (se*RUNX2* -791kb), which is activated in normal pDCs' on Page 7.

Reviewer #2 (Remarks to the Author):

In the revised manuscript, the authors have been largely responsive to the points and concerns raised in my original critique. Ideally, I would like to see them do some additional characterization of the immunophenotype of the tumors that arise in the mouse model. Other highly characteristic features of human BPDCN is expression of CD4, CD56, CD123, and TCL1, and failure to express lysozyme; some of these can be rapidly tested by flow cytometry and/or IHC.

Response: We appreciate the reviewer's comments on our revised manuscript. As the reviewer suggested additional characterization of the phenotype of murine BPDCN cells, we performed flow cytometry analysis of CD4, CD56, CD123/IL-3R and TCL1 on murine tumor cells. Reminiscent with previous studies that human pDCs, but not murine pDCs, expressed CD123/IL-3R (Hochrein H, et al. Hum Immunol. 2002 review), we found that the murine tumor cells did not express CD123 on their surfaces by flow cytometry (left figure) and showed lower level of *Il3ra* mRNA by RNA-sequencing than progenitors and mature pDCs (right figure). We also found that the murine tumor cells expressed neither CD4 nor CD56 on their surfaces, while the tumor cells mildly expressed TCL1 protein examined by intracellular flow-cytometry analysis as shown here (left figure), indicating the phenotypic difference in BPDCN cells between human and our murine model. We described this point in the discussion on Page 18. In addition, we examined whether the murine tumor cells expressed lysozyme by performing IHC, as the reviewer expected, the tumor cells did not express lysozyme as shown in Supplementary Figure S9 on Page 13.

REVIEWERS' COMMENTS:

Reviewer #2 (Remarks to the Author):

The authors have been responsive to my secondary comments; I am satisfied with their response.

March 12, 2019

RE: Revised version of a reference of *Nature Communications*: NCOMMS-18-11645B

REVIEWERS' COMMENTS:

Reviewer #2 (Remarks to the Author):

The authors have been responsive to my secondary comments; I am satisfied with their response.

Response: Thank you very much again for your time and efforts to improve our manuscript. We also like to thank Reviewer #1 for his/her time and help for improving our manuscript.

Goro Sashida, MD, PhD
International Research Center for Medical Sciences, Kumamoto University
Address: 2-2-1 Honjo, Chuo-ku, Kumamoto 860-0811, Japan
Phone: +81-96-373-6827
E-mail: sashidag@kumamoto-u.ac.jp

Motomi Osato, MD, PhD
Cancer Science Institute of Singapore, National University of Singapore
14 Medical Drive 117599, Singapore
E-mail: csimo@nus.edu.sg